# Atomic-scale observation of geometric reconstruction in a fluorine-intercalated infinite layer nickelate superlattice

Chao Yang [1,2] ✉, Roberto A. Ortiz [1,2], Hongguang Wang [1], Wilfried Sigle[1], Kelvin Anggara [1], Eva Benckiser [1], Bernhard Keimer [1] & Peter A. van Aken [1]

Anion doping offers immense potential for tailoring material properties, but precise control over anion incorporation remains challenging due to complex synthesis and limitations in dopant detection. This study investigates F-ion intercalation within an infinite-layer $NdNiO_{2+x}$/$SrTiO_3$ superlattice using a two-step process. We employ advanced four-dimensional scanning transmission electron microscopy (4D-STEM) coupled with electron energy loss spectroscopy (EELS) to map the F distribution and its impact on the atomic and electronic structure. Our observations reveal a fluorination-induced geometric reconstruction of the infinite layer structure, resulting in a more distorted orthorhombic phase compared to the pristine perovskite. F-ion are primarily located at apical polyhedral sites, with some basal sites occupation in localized regions, leading to the formation of two distinct domains. These domains reflect a competition between polyhedral distortion and Nd displacement at domain interfaces. Interestingly, we observe an anomalous structural distortion where basal site anions are displaced in the same direction as Nd atoms, potentially linked to the partial basal site F-ion occupation. This coexistence of diverse structural distortions signifies a locally disordered F-ion distribution with distinct configurations. These findings provide crucial insights into the mechanisms of anion doping at the atomic level, contributing to the design of materials with tailored functionalities.

Perovskite oxides have emerged as a versatile platform for exploring a multitude of fascinating phenomena by unlocking the intricate interplay between the crystal structure, electronic configuration, and processing conditions. These include colossal magnetoresistance[1], ferroelectricity and multiferroicity[2], and superconductivity[3]. A strategy to manipulate the properties of these materials lies in topochemical anion intercalation/exchange, where guest ions are introduced into the lattice while preserving the overall crystallographic framework. One particularly intriguing guest ion is fluorine (F), whose incorporation can dramatically alter material behavior[4–6]. For instance, F-ion exchange/intercalation in $NdNiO_3$ thin films induces a transformation

from a metallic to a highly insulating state, partially related to enhanced Coulomb repulsion in the Ni $3d$ orbitals, which is reversible after annealing in an oxygen atmosphere[1]. Furthermore, first-principles calculations show that these fluorinated films exhibit a stronger orthorhombic distortion compared to the pristine $NdNiO_3$ perovskite[7]. The calculations predict an ordered, anisotropic arrangement of F ions within the lattice[7], a feature crucial for understanding the properties of mixed anion compounds. Moreover, F-ion intercalation can lead to a rich variety of phenomena. For instance, Ruddlesden-Popper nickelate oxides ($n = 2$) exhibit a breaking of local inversion symmetry due to F-ion insertion, inducing a local

[1]Max Planck Institute for Solid State Research, Stuttgart, Germany. [2]These authors contributed equally: Chao Yang, Roberto A. Ortiz. ✉e-mail: c.yang@fkf.mpg.de

antiferroelectric state[2]. Similarly, the incorporation of F plays a crucial role in achieving superconductivity in $Sr_2CuO_2F_{2+x}$ cuprates[8-11]. However, a critical gap in our understanding remains: the precise influence of F-ion distribution on the atomic structure of perovskite oxyfluorides and the difference between the O and F positions has not been directly observed experimentally.

Beyond anion intercalation, deintercalation via topochemical reactions offers another strategy to manipulate the properties of perovskites. This approach modifies the atomic and electronic structures of the host material by removing specific anions. A prime example lies in the synthesis of infinite layer structures from ferrites[12,13] and nickelates[14-17]. Here, reducing agents like $CaH_2$ are employed to remove apical oxygen from the octahedra in their corresponding perovskite phases[18]. The resulting infinite layer structures have garnered significant interest, particularly due to the recent discovery of superconductivity in $Nd_{0.8}Sr_{0.2}NiO_2$ films[3]. This has spurred intense research to elucidate the similarities and differences between nickelates and cuprates in terms of magnetic structure[19,20], superconducting behavior[21,22], and charge ordering[23-25]. Understanding these aspects is crucial for advancing the theory of high-temperature superconductivity. Resonant X-ray scattering experiments revealed a commensurate charge density wave order at a wave vector of (0.333, 0) reciprocal lattice units in the infinite layer structure, suggesting a distinct multiorbital character compared to cuprates[23]. However, a combination of X-ray scattering and atomic-scale observations has challenged this interpretation, suggesting the $3a_0$ superlattice peak might originate from an intermediate nickelate phase with residual apical oxygen ordering[26]. This underscores the critical role of precise control over synthesis and the need for atomic-scale interrogation of local atomic and electronic structures to fully comprehend the physical properties of these complex systems.

The intricate nature of topochemical reactions necessitates the exploration of alternative strategies for achieving precise control over anion ordering in oxyfluorides. A promising approach may lie in the combination of oxygen deintercalation and subsequent fluorine intercalation. This sequential approach offers the potential to engineer specific oxygen/fluorine arrangements, paving the way for a controllable manipulation of functional properties in these materials. In particular, the partial substitution of F for O can induce an electronic configuration of Ni $3d^{8.8}$ in the nickelate system, which has been theoretically linked to superconductivity. However, to fully exploit the potential of F-ion intercalation, a comprehensive understanding of the O/F configuration, its influence on atomic-scale structural geometry, and the resulting electronic structure is crucial.

In this work, we address this critical knowledge gap by investigating F-ion intercalation in infinite layer nickelates within an $8NdNiO_2/4SrTiO_3$ superlattice structure, building upon our prior work on the synthesis of this material[16]. To achieve atomic-level resolution of the oxygen and fluorine distribution, we employed advanced microscopy techniques – specifically, aberration-corrected 4D-STEM coupled with EELS. Furthermore, we utilized the integrated center of mass (iCoM) imaging technique to map the anion positions with sub-angstrom precision. This multifaceted approach revealed a fascinating geometric reconstruction in the nickelate layer induced by the F-ion incorporation. We will subsequently discuss the implications of the observed F-ion distribution on the structural distortions and electronic structure, drawing insights from density functional theory (DFT) calculations.

## Results and discussion

Figure 1a schematically illustrates the two-step synthesis employed to create a fluorine-intercalated infinite layer nickelate. The first step involves the formation of the infinite layer structure within the $8NdNiO_2/4SrTiO_3$ superlattice through a topochemical reduction process utilizing $CaH_2$ as a reducing agent. This step entails

deintercalation, where apical oxygen is predominantly removed from the oxygen octahedra within the $NdNiO_3$ layer (detailed in Fig. S1 and our previous work[16]). However, a recent study of a larger volume fraction of the sample by linear dichroic resonant x-ray reflectometry shows an approximately 30% fraction of nickelate where the basal oxygen (oxygen atoms in the equatorial plane of the octahedra) is removed instead of the apical oxygen[27]. A key observation is that the infinite layer structure achieves stability within the confined thickness of the superlattice or when employing a capping layer[16]. Subsequently, F atoms readily intercalate into the vacant apical positions within the infinite layer structure, forming a $NiO_xF_y$ polyhedron. The large field of view high-angle annular dark-field (HAADF) images in Fig. S2 show the microstructures of $8NdNiO_2F_x/4SrTiO_3$ and $8NdNiO_2/4SrTiO_3$ superlattices.

Figure 1b presents a HAADF image of the atomic structure within the $NdNiO_2$ and $SrTiO_3$ layers of the superlattice. Compared to bulk $NdNiO_3$ (out-of-plane spacing of 3.86 Å), the $NdNiO_2$ layer exhibits a significant reduction in this spacing to around 3.2 Å. Additionally, the characteristic zigzag arrangement of A site cations is absent in the infinite layer structure. The corresponding maps and line profiles for the zigzag angle remain consistent across both the $SrTiO_3$ and $NdNiO_2$ layers, strongly suggesting successful deintercalation of the apical oxygen anions. In contrast, Fig. 1c showcases the HAADF image of the structure after fluorine intercalation into the $NdNiO_2$ layer. This incorporation of fluorine induces substantial structural modifications, evident from the expansion of the out-of-plane lattice spacing and the reappearance of a zigzag cation arrangement. The corresponding zigzag angle map on the right of Fig. 1c reveals an inhomogeneous distribution of the zigzag arrangement of Nd atoms across different F-intercalated $NdNiO_2$ layers.

Furthermore, the displacement of Nd atoms within the same F-intercalated $NdNiO_2$ layer exhibits inconsistencies. The zigzag angle profile on the right side of Fig. 1c reveals that the Nd atoms in the inner layer reach a maximum zigzag angle of around 9°, signifying a substantial rotation of their octahedra. This rotation is slightly stronger compared to bulk $NdNiO_3$ (-8.5°)[28]. Interestingly, the angle sharply decreases to about 2° at the interfaces with the neighboring layers, suggesting a return to octahedral coupling at these boundaries. Figure 1e highlights regions within the F-intercalated $NdNiO_2$ layer that exhibit an apparent zigzag arrangement of Nd atoms, absent in the reduced structure (Fig. 1d). These observations, along with the broader overview provided in Figure S4, point towards the formation of domains within the $NdNiO_xF_y$ layers characterized by different zigzag arrangements of Nd atoms. Specifically, the domain with zigzag arrangement of Nd atoms can be formed by the 90° rotation of the domain without zigzag arrangement of Nd atoms. This inhomogeneous distribution suggests that strong structural coupling might occur not only at the interfaces between layers but also within the domain wall regions themselves.

While conventional HAADF microscopy struggles with oxygen imaging, the latest 4D-STEM technique offers a solution. By reconstructing iCoM images, we achieved high-resolution visualization of the anion sublattice, enabling precise determination of structural distortions[29]. Figure 2a showcases the reconstructed ADF image from the 4D-STEM data, revealing the high-quality atomic structure of the $8NdNiO_xF_y/4SrTiO_3$ superlattice. The corresponding iCoM image in Fig. 2b clearly depicts the anion sublattice, highlighted by the prominent zigzag arrangement of Nd atoms and a pronounced octahedral distortion within the regions delineated by yellow dashed boxes. To quantify this distortion, the zigzag angle map of Nd atoms, shown in Fig. 2c, utilizes the Sr atoms' map as a reference. The averaged zigzag angle line profile in Fig. 2d reveals significant variations, with the highest angle reaching -12° and gradually decreasing to around zero within the $SrTiO_3$ layer. The zigzag angle in the $NdNiO_xF_y$ layer significantly surpasses that observed in perovskite $NdNiO_3$, signifying a

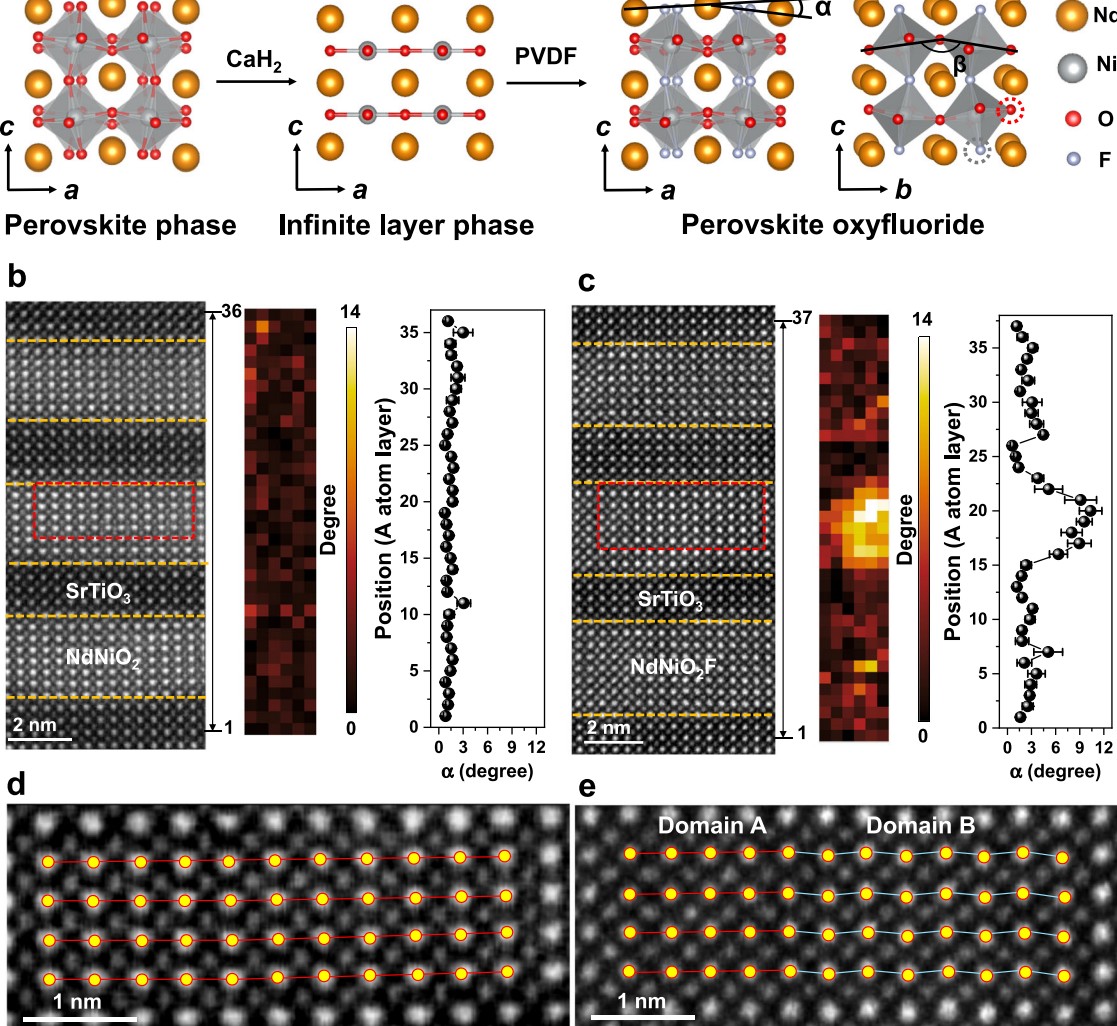

**Fig. 1 | Synthesis of the fluorine-intercalated infinite layer nickelates and the formation of different domains in the oxyfluoride NdNiO₂F. a** Structural models of the perovskite phase, the infinite layer phase and the oxyfluoride phase. α is the zigzag angle α between the A-site cations, specifically the Nd and Sr ions. β is for the Ni-O-Ni angle. The red and gray dashed circles indicate the potential for occupancy of the O and F atoms at the basal and apical sites, respectively. **b** HAADF image of the 8NdNiO₂/4SrTiO₃ superlattice, a map of the zigzag angle α, and a line profile of the zigzag angle α between the A-site cations (A: Nd and Sr). **c** HAADF image of the 8NdNiO₂F/4SrTiO₃ superlattice, a map of the zigzag angle α, and a line profile of the zigzag angle α between the A-site cations (A: Nd and Sr). Thirteen Nd atom points were used to calculate 6 zigzag angles per row, resulting in a 6 × 36 map in (**b, c**). The exact regions used to calculate the zigzag angles in **b, c** are shown in Fig. S3. The error bars in **b, c** are derived from the Gaussian fit using the Atomap package. **d** A magnified HAADF image of the region marked by the red dashed box in **b**. **e** A magnified HAADF image of the region marked by the red dashed box in **c** showing two domains. In **d, e**, the yellow dots and the red and blue lines are guides for the eye, indicating A-site cation positions, straight and zigzag cation arrangements, respectively.

strong structural distortion induced by F-ion intercalation. We further quantified the in-plane B-O-B angle (B: Ni and Ti) in Fig. 2e. A strong correlation emerges: regions with a lower B-O-B angle in the NdNiOₓFᵧ layer coincide with regions exhibiting a high zigzag angle. This suggests an unusual in-plane Ni-O-Ni distortion due to F intercalation, a phenomenon not previously reported, a likely related to the two-step synthesis approach used in this work.

Figure 2f presents a magnified ADF image highlighting the pronounced zigzag arrangement of Nd atoms. In principle, the corresponding structural model would be expected to be the orthorhombic structure depicted in Fig. 2g, where two neighboring oxygen atoms reside at each basal site from this viewing direction. However, due to the limitations of TEM resolution, these adjacent oxygen atoms are difficult to distinguish, resulting in a near-zero in-plane Ni-O-Ni angle in this projection. Remarkably, the magnified iCoM image in Fig. 2h unveils an intriguing structural distortion where the basal oxygen simultaneously shifts in the same direction as the Nd atoms, as

illustrated by the green polyhedral model in Fig. 2i. Furthermore, the iCoM image in Fig. S6 exhibits a comparable structural distortion consistent with Fig. 2i, alongside a significant polyhedral rotation in a region lacking Nd displacement. These iCoM images show the significant differences in atomic arrangement between the two observed domains within the anion sublattice.

To definitively confirm F-ion intercalation within the NdNiO₂ layer, we employed high-resolution STEM-EELS to map the elemental distribution across the interface in the NdNiO₂/SrTiO₃ superlattice (Fig. 3). The HAADF image in Fig. 3a demarcates the region analyzed by EELS, where the distinct contrast readily differentiates the NdNiO₂ and SrTiO₃ layers. The corresponding elemental maps in Fig. 3b, acquired at the Nd M₄,₅, Ni L₂,₃, Sr L₂,₃, Ti L₂,₃, O K, and F K edges, visualize the atomic-scale distribution of constituent elements. It is noteworthy that a prominent F signal is present within the NdNiO₂ layer and absent in the SrTiO₃ layer.

Figure 3c depicts the row-averaged line profile of the oxygen signal intensity extracted from the region marked in Fig. 3a. The

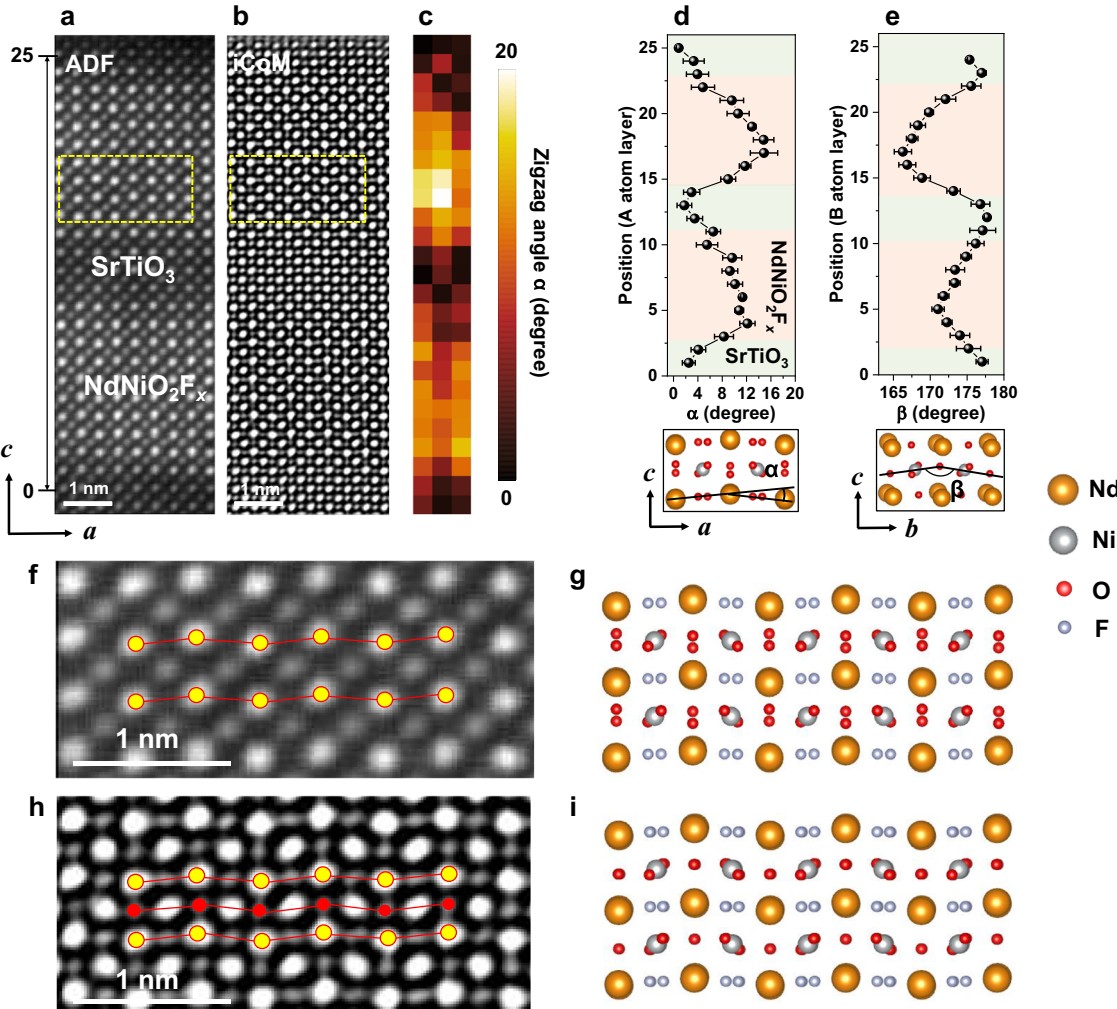

**Fig. 2 | Atomic distortion in the oxyfluoride NdNiO$_2$F$_x$. a** ADF and **b** iCoM images of the 8NdNiO$_2$F/4SrTiO$_3$ superlattice reconstructed from the 4D-STEM data set. Quantified zigzag angle α **c** map and **d** line profile of the A-site cations (A: Nd, and Sr) in the region marked from the 1st to the 25th unit cell in **a. e** B-C-B (B: Ni, and Ti, C: basal anions) angle plot of the same region in **d**. The green and red shaded areas in **d**, **e** correspond to SrTiO$_3$ and NdNiO$_2$F$_x$, respectively. Seven Nd atom points were used to calculate three zigzag angles per row, forming a 3 × 25 map in **c**. The exact region for the calculation of zigzag angles α and B-C-B angles β in **c**–**e** is

shown in Fig. S5. The error bars in **d**, **e** have been derived from the Gaussian fit using the Atomap package. **f** Enlarged ADF image of the region marked by the yellow dashed box in **a** and the estimated structure model in **g. h** Enlarged iCoM image of the region marked by the yellow dashed box in **b** and the estimated structure model in **i**. In **f**, **h**, the yellow and red dots and red lines are guides for the eye, indicating the A-site cation positions, anion positions, and zigzag atomic arrangements, respectively.

gradual decrease in oxygen intensity from the interface towards the inner nickelate layer reflects the variation in oxygen concentration. Conversely, the F-signal intensity in Fig. 3d exhibits a gradual increase moving inwards from the interface. This observation suggests a higher concentration of F ions in the inner NdNiO$_2$F layer compared to the interfaces. This difference can be attributed to the presence of residual oxygen at the interfaces within our NdNiO$_2$/SrTiO$_3$ superlattice, as previously reported[16].

Furthermore, to estimate the oxidation state of Ni atoms in the NdNiO$_2$F layer, we quantified the Ni L$_3$/L$_2$ ratio (Fig. 3e). The corresponding spectra of O K and Ni L edges are shown in Fig. S7. The Ni valence state within the inner nickelate layer fluctuates around 2+, with occasional slight deviations below 2+. This finding suggests a possible composition of either NdNiO$_2$F or NdNiO$_2$F$_x$ (where $x < 1$). In contrast, the Ni valence state near the interfaces exhibits an intermediate state between Ni$^{2+}$ and Ni$^{3+}$. This observation can be attributed to the influence of residual oxygen at the interface[16], potentially leading to an NdNiO$_{2.5}$F$_{0.5}$ configuration.

Theoretical calculations on various oxyfluorides predict anion ordering based on energetic stability. These calculations typically suggest F-ion intercalation at either apical or basal sites of the oxygen octahedra, or in interstitial sites within Ruddlesden-Popper structures[7,30–34]. Given our two-step synthesis approach, F-ion intercalation within the apical sites was anticipated, leading to an ordered O/F configuration. To determine the specific distribution of F-ions within the infinite layer structure, we quantified EELS measurements at specific positions (Fig. 4). The HAADF image in Fig. 4a, b depicts the atomic structure chosen for EELS analysis. Red and yellow dots in Fig. 4b mark the targeted apical and basal positions within the polyhedra, respectively. To minimize the influence of signal mixing, EELS measurements were performed in a region with a t/λ value of ~0.2 (see Fig. S8), corresponding to a sample thickness of approximately 15 nm. Since beam scattering at this thickness can introduce a degree of signal mixing from adjacent columns into the EELS results, we performed the atomically resolved EELS simulation shown in Fig. S9. There is ~6% of the O signal at the F site and ~16% of the F signal at the O site, where the

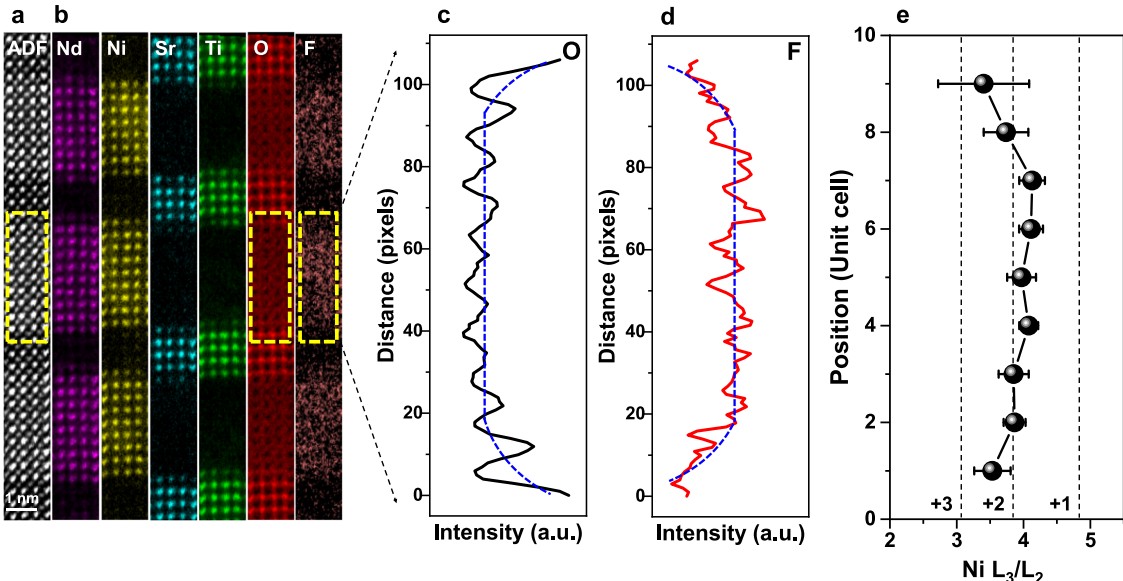

**Fig. 3 | Elemental distribution, composition, and valence state in the oxy-fluoride NdNiO$_2$F superlattice. a** ADF image for the EELS measurement of the 8NdNiO$_2$F/4SrTiO$_3$ superlattice. **b** The corresponding element distributions: Nd (purple), Ni (yellow), Sr (blue), Ti (green), O (red) and F (orange). **c** Intensity line profile of the O K edge signal extracted from the region marked by the yellow dashed box in **a**. **d** Intensity line profile of the F K edge signal extracted from the region marked by the red dashed box in **a**. In **c**, **d**, the blue dashed lines are guides for the eye. **e** The calculated Ni L$_3$/L$_2$ intensity ratio in the nickelate layer. The dashed lines indicate the reference values of the Ni L$_3$/L$_2$ intensity ratio for different Ni valence states, which are derived from NdNiO$_3$ for Ni$^{3+}$[52], NiO for Ni$^{2+}$[53], and NdNiO$_2$ for Ni$^{+}$[16]. The error bars in **e** are computed from multiple data sets.

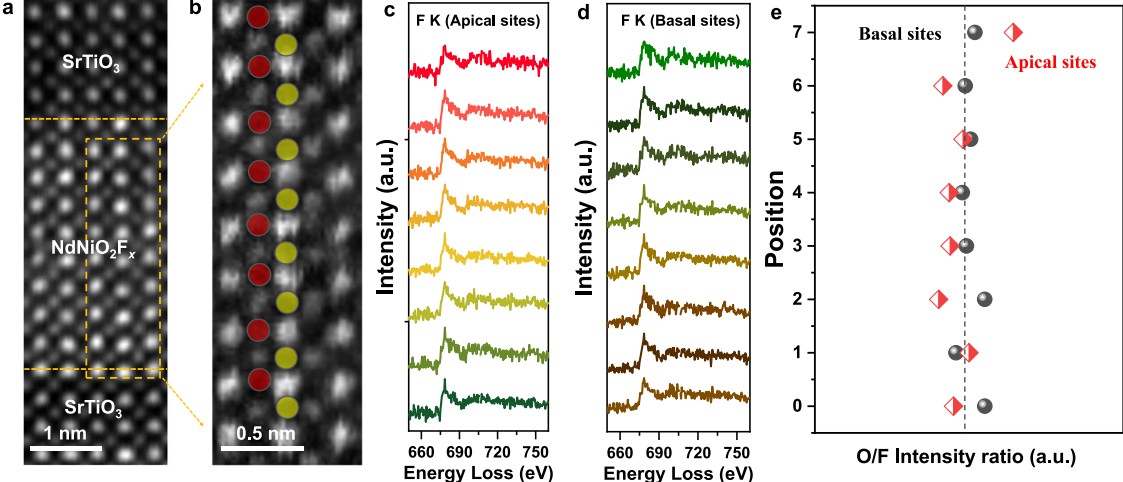

**Fig. 4 | Detected F-signal at both basal and apical sites and its effect on the atomic structure distortion. a** Overview and **b** enlarged HAADF images of the region marked by yellow dashed box in **a** for EELS measurements of the NdNiO$_2$F$_x$ inner layer. The yellow dashed lines in **a** mark the SrTiO$_3$/NdNiO$_2$F$_x$ interfaces. Red and yellow spots in **b** mark the apical and basal sites, respectively. **c** The F signal extracted from the red spots in **b**. **d** The F signal extracted from the yellow spots in **b**. **e** The relative O/F signal intensity ratio at the apical and basal sites.

regions for calculation are marked in the composite map in Fig. S9f. To determine the position of the intercalated F-ions, we extracted EELS spectra corresponding to the F-K edge at both the apical and basal positions, as shown in Fig. 4c, d. Both positions exhibit a significant F signal, with approximately the same F/O signal intensity ratio at both apical and basal positions in the inner nickelate layer (see Fig. 4e), indicating the F intercalation locally at both apical and basal sites.

To elucidate the F/O configuration and its impact on structural distortion, we employed DFT calculations in conjunction with analysis of the observed distortions. The HAADF image in Fig. S10c reveals a significantly elliptical Nd column. This observation is in good agreement with the relaxed structural model in Fig. S10d and the simulated HAADF image in Fig. S11c, which shows F-ion intercalation at the apical sites. The elliptical shape of the Nd column arises from a viewing direction mismatch, leading to a more pronounced effect compared to perovskite nickelate (see Fig. S12). An alternative structural model is presented in Fig. S10e, featuring a 90° rotation compared to Fig. S10d. This rotation results in a distinct atomic arrangement with a different zigzag distortion of the Nd atoms. Interestingly, in localized regions, we observe adjacent Nd columns with elliptical shapes oriented perpendicular to each other, as shown in Fig. S10f. This experimental finding is in agreement with the DFT

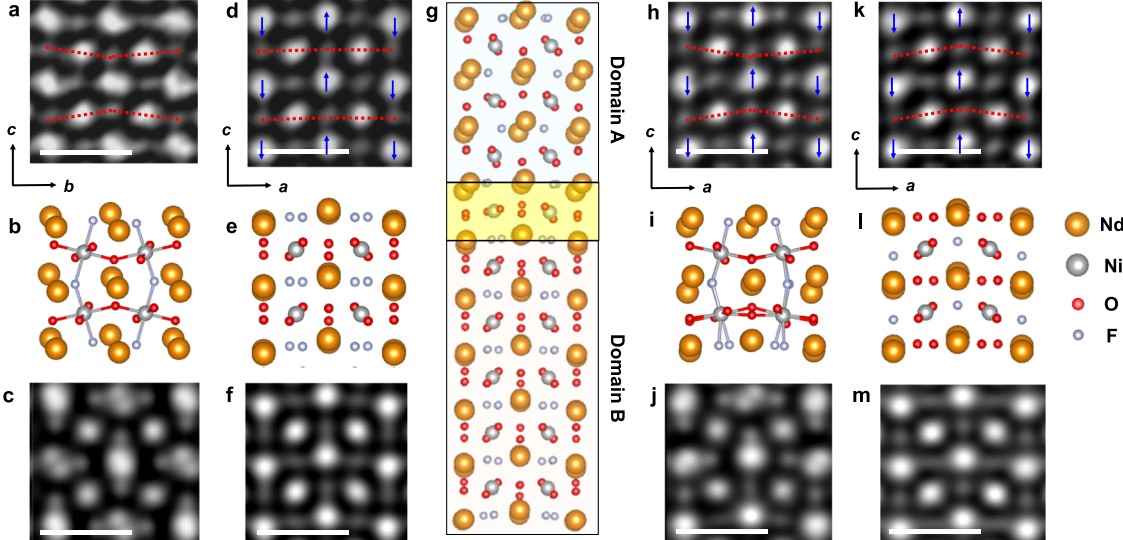

**Fig. 5 | Comparison between the experimentally observed structure distortion and the relaxed structure model from DFT calculations. a** Reconstructed iCoM image of domain A with clear polyhedral rotation and **b** the relaxed structure model with F intercalation at apical sites. The red dashed lines in **a** mark the anion sublattice distortion. **c** A simulated iCoM image of the structure in **b**. **d** Reconstructed iCoM image of domain B showing a clear zigzag arrangement of Nd atoms and **e** the relaxed structure model with F intercalation at apical sites. **f** A simulated iCoM image of the structure in **e**. The blue arrows indicate the displacement direction of the Nd atoms and the red dashed line marks the anion sublattice without distortion in **d**. **g** A relaxed structural model with two connected domains. Domain B is formed by a 90° rotation of domain A. **h** Reconstructed iCoM image of the domain interface and **i** the enlarged structural model. **j** A simulated iCoM image of the structure in **i**. The blue arrows indicate the displacement direction of the Nd atoms and the red dashed line marks the anion sublattice distortion in **h**. **k** Reconstructed iCoM image of the region with abnormal structure distortion and **l** the structure model from the DFT calculations with partial intercalation of F-ions at the basal sites. **m** A simulated iCoM image of the structure in **l**. The blue arrows indicate the displacement direction of the Nd atoms and the red dashed line marks the anion sublattice distortion in **k**. The white lines of the scale bar represent a length of 0.5 nm in the figures.

calculated model in Fig. S10g and the simulated HAADF image in Fig. S11f, suggesting F-ion intercalation at the basal sites. This observation further corroborates the EELS measurements presented earlier. A 90° rotation of the viewing direction in Fig. S10h, relative to Fig. S10g, reveals a similar zigzag distortion of the Nd atoms as in Fig. S10e. However, a slightly stronger polyhedral distortion is evident, manifested by a greater separation of the adjacent anions at both apical and basal sites.

We now turn our attention to the anion sublattice within the F-intercalated $NdNiO_2$ layer. The HAADF overview image in Fig. S13 shows the region where we cropped the local regions in Fig. 5. The iCoM image in Fig. 5a depicts a local region exhibiting a typical orthorhombic structure with evident octahedral rotation. Additionally, a slight mismatch of the Nd columns is observed from this viewing direction. In particular, the intercalation of F ions at apical sites, as shown in the relaxed structure model under $SrTiO_3$ strain (Fig. 5b) and the corresponding simulated iCoM image (Fig. 5c), leads to an increased Nd mismatch, consistent with the elliptical Nd columns observed in the HAADF image (Fig. S10c). A 90° rotation of the viewing direction (from [100] to [010]) in Fig. 5d reveals a distinct atomic arrangement. Here, a pronounced zigzag displacement of Nd atoms is evident, while no visible polyhedral distortion is observed. This finding is in good agreement with the relaxed structure model shown in Fig. 5e and the simulated iCoM image in Fig. 5f. Interestingly, the presence of two structural configurations within a single viewing direction (Figs. 1e and S4) suggests the coexistence of widely distributed domains A and B in the F-intercalated $NdNiO_2$ layers. Domain A is the main configuration in the $NdNiO_xF_y$ layers. Domain B, with obvious Nd zigzag arrangement, is randomly distributed within a few nanometers. Figure 5g showcases a relaxed structure model depicting these two domains, highlighting the clear structural coupling at the domain interface marked by the yellow box.

The iCoM image in Fig. 5h provides compelling experimental evidence for this structural coupling at the domain interface. Here, we observe competition between the Nd ion displacement and the $NiO_xF_y$ polyhedral distortions, reflecting the calculated relaxed structure in Fig. 5i and the simulated iCoM image in Fig. 5j. Furthermore, the iCoM images in Figs. 5k and 2h reveal an intriguing structural distortion where the basal anion displacement direction aligns with that of the Nd atoms. This observation points towards a configuration distinct from the F-ion intercalation at apical sites depicted in Fig. 5d–f. As shown in Fig. S14b, F intercalation at basal positions can lead to a slightly greater polyhedral distortion compared to intercalation at apical positions. This distortion can induce a shift in one of the basal F-atoms, bringing it closer to the Nd atoms. This phenomenon explains the weak contrast observed between Nd columns and basal F ions in Fig. S14a (marked by red arrows). For regions with no visible contrast between Nd and basal anions in Fig. 5k and S14c (marked by yellow arrows), we propose a composition of $NdNiO_2F_{0.5}$. This model includes partial F-atom intercalation within the $NdNiO_2$ layer, as shown in the relaxed structure model (Fig. 5l) and the simulated iCoM image (Fig. 5m). This configuration effectively explains the observed lack of contrast between Nd and basal anions.

The interplay of anion intercalation and deintercalation in nickelates triggers significant structural modifications. In the reduced $8NdNiO_x/4SrTiO_3$ superlattice sample, residual oxygen persists at the interfaces and inner layers despite the stabilization of the infinite layer structure. This residual oxygen impacts F-ion incorporation within the reduced nickelate. Chemical composition measurements and Ni valence quantification suggest a configuration near the interfaces resembling $NdNiO_{2.5}F_{0.5}$ with a polyhedral $NiO_5F$ structure. The coupling between $TiO_6$ octahedra and $NiO_5F$ polyhedra induces subtle Nd atom displacements alongside polyhedral rotations within the interfacial nickelate layers.

In contrast, F-ion intercalation within the infinite layer structure leads to a substantial geometric reconstruction, resulting in an orthorhombic structure akin to perovskite nickelate, but with a more pronounced distortion. The observed gradient structural changes from the interface to the inner layer are mainly influenced by the atomically non-sharp interface, which is consistent with previous findings in perovskite superlattices[35]. Due to the two-step fluorination process, F ions are primarily intercalated at the apical sites of the polyhedron, with partial intercalation at the basal sites. We have distinguished the F-ion intercalation positions (basal vs. apical) based on the varying cation displacements observed in HAADF images, which aligns with the EELS-determined chemical composition. Intriguingly, two widely distributed domains form within the nickelate layers. In the vicinity of the domain interface region, there is a competition arises between the Nd displacement and the $NiO_xF_y$ polyhedral rotations. Additionally, a structural distortion is observed within domain B, potentially arising from the partial intercalation of F ions at basal sites and exhibiting an O/F ordered arrangement according to DFT calculations.

In addition, DFT calculations including the on-site Coulomb interaction (U) elucidate the influence of the F-ion intercalation-induced perturbations observed in our experiments on the electronic structure of the nickelate. F-ion intercalation drives the Ni valence state toward 2+ according to the EELS results in Fig. 3e, thereby inducing a distinct band gap[1], as reflected in the density of states (DOS) for the $NdNiO_2F$ configuration (Fig. S15). There is no clear difference in the band gap in the case of F intercalation in apical or basal sites. With a lower F intercalation of the structure model in Fig. S15e, the band gap shows a significant decrease. In fluorinated samples with different F concentrations, the band gap values vary in the range of 0.84 eV to 2.1 eV, according to the optical measurements[1]. The DFT calculations of our samples show comparable change in this system. Resistance measurements of the fluorinated superlattice show highly insulating behavior, so that it was not possible to measure its temperature dependence. In addition, the valence band is dominated by hybridized Ni/O bands, with the F band positioned far from the Fermi level in the $NdNiO_2F$ structure. As shown in Fig. S15 for various F-intercalation configurations, the band gap is primarily influenced by the overall F-ion concentration, with a minor dependence on the specific F-ion position (basal vs. apical). This can be attributed to the modification of electron occupancy in the Ni orbitals upon F-ion intercalation within the infinite layer structure. The on-site Coulomb interaction subsequently suppresses electron hopping between neighboring Ni sites in the $NdNiO_2F$ structure[7].

In summary, our findings demonstrate that the infinite layer structure serves as a compelling model system for tailoring anion intercalation. We present direct observations of the geometric reconstruction process triggered by F-ion intercalation within the nickelate's infinite layer structure. Due to the comparable ionic radii of $F^-$ and $O^{2-}$, F-ion incorporation leads to the formation of a sixfold-coordinated orthorhombic structure akin to perovskite nickelates, but exhibiting a more pronounced distortion. Residual oxygen persisting at the interfaces in the reduced superlattice sample can promote the formation of a $NiO_5F$ configuration upon fluorination. Furthermore, we reveal the formation and widespread distribution of two distinct domains within the nickelate layers. These domains exhibit significant coupling at the domain wall, leading to a competition between Nd displacement and the $NiO_xF_y$ polyhedral distortion. We also observed F-ion intercalation at basal sites in localized regions, evidenced by the distinct elliptical direction of adjacent Nd columns. This observation is further corroborated by high-resolution EELS measurements. Additionally, a structural distortion was identified, characterized by the alignment of Nd atom and basal anion displacements. This phenomenon, supported by DFT calculations, suggests partial F-ion intercalation at basal sites, providing evidence for anion mobility during the fluorination process.

Our findings highlight the $NdNiO_2F$ configuration as the dominant phase within the nickelate layer. Here, the on-site Coulomb repulsion between neighboring $Ni^{2+}$ cations effectively suppresses electron transport. Since F-ion intercalation primarily influences the Ni electronic configuration, precise control over this process within the infinite layer nickelate system appears critical for achieving the desired Ni $3d^{8.8}$ configuration, a potential route towards superconductivity[3,36]. Furthermore, previous reports suggest that the electronic structure modifications induced by F-ion incorporation in nickelates can be reversed by subsequent oxygen intercalation through annealing in an oxygen atmosphere[1]. This opens avenues for the development of resistance switching materials. Our work also sheds light on the strategic application of combined anion intercalation and deintercalation for investigating the ordered arrangement of anions in related systems. The $Ni^{2+}$ in the F-intercalated infinite layer structure underscores the significant influence of ordered/disordered anion arrangements on the magnetic structure of nickelates, as previously reported[30,32]. This dependence highlights the intricate interplay between anion distribution and magnetic properties, paving the way for the tailored manipulation of magnetic behavior through controlled manipulation of the anion arrangement. In conclusion, elucidating the effects of F-ion intercalation on the atomic and electronic structures within the infinite layer structure system holds broader implications for perovskite oxyfluoride materials, where both polyhedral distortion and anion order/disorder play prominent roles.

## Methods

### Materials and sample preparation

The $NdNiO_3/SrTiO_3$ superlattice samples were epitaxially grown on a (001)-oriented single crystal $SrTiO_3$ substrate by pulsed laser deposition (PLD) technique under the same conditions described in refs. 16,27,37,38, using a KrF excimer laser with 2-Hz pulse rate and 1.6-J/cm$^2$ energy density. The superlattice film was deposited in 0.5-mbar oxygen atmosphere at 730 °C and then annealed in 1-bar oxygen atmosphere at 690 °C for 30 min. Subsequently, the topotactical reduction was performed with $CaH_2$ powder inside a vacuum sealed Pyrex glass tube at a temperature of 280 °C for aproximately four and a half days. The preparetion of the sample with the $CaH_2$ powder and the glass tube was performed inside an Ar-filled glove box to prevent any oxygen contaminants[3,16,37]. The fluorination process is then performed using polyvinylidene fluoride (PVDF)[30,34]. The process took place in a Tube Furnace which was filled with a constant Ar flow before and during the chemical reaction. The heater was ramped at approximately 5 °C/min until the furnace reached a stable temperature of 350 °C. Then, the process continued for one hour and the heater was turned off. The system was allowed to cool down to room temperature which usually took between 4 and 5 hours. After this, the sample was removed and cleaned with Acetanol. The X-ray diffraction (XRD) results in Fig. S16 show the phase transitions induced by fluorination in both perovskite and infinite-layer phase nickelate superlattice samples. The TEM lamellae were prepared using a focused ion beam (FIB Scios, FEI) in a high-vacuum environment[39]. To mitigate beam-induced charging effects during sample preparation, a thin layer of approximately 6 nm of carbon was deposited on the sample surface using a high-vacuum sputter coater (EM ACE 600, Leica). A Fischione NanoMill® TEM sample preparation system was used to improve the quality of the TEM lamellae through low-energy milling and high-vacuum cleaning.

### STEM imaging and STEM-EELS simulation

The STEM studies were performed at 200 kV using a JEOL JEM-ARM200F microscope (JEOL Co. Ltd.) equipped with a DCOR probe corrector and a Gatan GIF Quantum ERS K2 spectrometer. For STEM imaging, a condenser aperture of 30 μm was used, with a corresponding convergence semi-angle of 20.4 mrad. We use a collection

semi-angle between 83 and 205 mrad for HAADF imaging, and 85 μs for EELS measurements. A dispersion of 0.5 eV/channel with an energy resolution of about 1 eV was used to acquire the EELS spectra. A relative thickness map ($t/\lambda$) is expressed in units of the inelastic mean free path $\lambda$. EELS data analysis was performed using Digitalmicrograph software. For the white line ratio, we first denoised the EELS data using principal component analysis (PCA) in DigitalMicrograph. The background was then removed using a power-law model, and the cross-sections were calculated based on the Hartree-Slater model. Finally, the white line ratio was calculated from the extracted integral $L_3$ and $L_2$ white line intensities. The white line ratio was used to estimate the valence of Ni[40–44]. The 4D-STEM data set was acquired using a Merlin pixelated detector (256 × 256 pixels, Quantum Detectors) in 1-bit mode, with continuous read/write at a pixel time of 48 μs. Post-processing of the 4D-STEM dataset is based on the Python libraries of py4dstem[45] and fpd[46]. The calculation of zigzag analge is based on the Python library of Atomap[47]. The atomically resolved STEM-EELS simulations were performed using an open-source Python package from abTEM[48]. A 4 × 4 × 40 unit cell superstructure, corresponding to the same sample thickness as we studied experimentally, was used for the calculations. The atomic potential was taken from the independent atom model potential[48]. The beam energy, convergence angle, and collection angle in the simulations are the same as those used in the experiment.

## DFT calculations

First-principles DFT calculations were performed to investigate the structure distortion and the DOS variation of the F-ion intercalation in the infinite layer nickelate in the $NdNiO_2/SrTiO_3$ superlattice. The calculations used the Generalized Gradient Approximation (GGA) and the Perdew-Burke-Ernzerhof (PBE) functional for exchange correlation, implemented in the Vienna Ab initio Simulation Package (VASP)[49,50]. The plane-wave cutoff energy of 520 eV was used. The DFT + U method used a Hubbard U parameter of 4.0 eV for the Ni atoms. For the structure optimization, the maximum ionic force is set to 0.01 eV/Å, the self-consistent convergence of the total energy is $5 \times 10^{-7}$ eV/atom and the maximum ionic displacement is set to 0.5 Å. The VASP calculated data were analyzed using the VASPKIT code[51].

## Reporting summary

Further information on research design is available in the Nature Portfolio Reporting Summary linked to this article.

## Data availability

The data generated or analyzed in this study are accessible from the corresponding author. Source data are provided with this paper.

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

## Acknowledgements

This project has received funding from the European Union's Horizon 2020 research and innovation programme under Grant Agreement No. 823717-ESTEEM3. The authors are thankful to Dr. Y. Wang for the support of 4D-STEM data processing, Dr. T. Heil for the support of merlin software, Dr. K. Kern, Dr. J. Tatchen, Dr. A. Schuhmacher, and X. J. Zhang for the support of the supercomputer software, K. Hahn and P. Kopold for TEM support, and Dr. D. S. Weng for the discussion of DFT calculations. The authors would like to thank Roland Eger for his help in setting up the fluorination setup.

## Author contributions

C.Y. and R.A.O. conceived the project. C.Y., H.G.W., W.S., and P.A.v.A. conducted the STEM measurements and related data analysis. R.A.O. grew the samples and performed the totapical reduction and fluorination. P.A.v.A., E.B., and B.K. supervised this work. C.Y. did the simulations and calculations. K.A. provided insight of the DFT results. C.Y. wrote the paper with contributions from all authors. All authors contributed to the discussion and comments.

## Funding

## Competing interests

The authors declare no competing interests.
