## [Transparent Peer Review file · Nature Communications]

Atomic-scale observation of geometric reconstruction in a fluorine-intercalated infinite layer nickelate superlattice

Corresponding Author: Dr Chao Yang

Version 0:

Reviewer comments:

Reviewer #1

(Remarks to the Author)

In this manuscript, the authors use detailed atomic-scale scanning transmission electron microscopy imaging and spectroscopy to reveal subtle localized structure inhomogeneities in the F-intercalated layers of a SrTiO₃/infinite-layer nickelate superlattice. 4D-STEM techniques and electron energy loss spectroscopy show how fluorine intercalates at different anion lattice sites, revealing in some localized regions how this leads to unexpected structural distortions. The experimental results are further supported by functional-density-theory-based calculations.

This study of oxyfluorides is particularly timely, with numerous research efforts currently directed at exploiting the emerging properties of these intriguing materials, but for which precise control of chemistry and structure is required. The results will thus be of interest to a wide research community beyond expert electron microscopists.

The experimental data is excellent and the careful and thorough processing offers unique insights into a system that is set apart by its two-step growth process – some of the O anions being stripped out of the lattice prior to F re-introduction. The authors show convincingly how this leads to unexpected localized geometric distortions of the lattice, with subtle chemical/structural interplay. In turn, these may have important consequences on the properties of the films.

Nevertheless, a number of reasonably minor issues should be addressed before this work is accepted for publication, as detailed below.

1. The authors rely on the iCOM technique to study the structural displacement of light-element atomic columns (O and F). The observation of several distinct structures in separate domains and from the same images/datasets certainly lends credence to the interpretation in terms of octahedral tilts and distortions depending on the presence of O or F at apical or basal sites. Nevertheless, as with all phase-contrast-based imaging (in a broad sense – iCOM is of course very specific), channeling and other phase effects may affect the image formation, especially in the thicker areas of the sample and given the presence of heavy cation columns such as Nd. Given that the authors have DFT-optimized models, simulations of expected iCOM contrast would offer a further welcome validation of the image interpretation, and they should consider adding these simulations to a revised manuscript.

2. One additional worry is local tilt along the beam direction. Figure S2 suggests the domains are separated by a few to 10s of nm, a distance over which the local crystal tilt could possibly change enough to affect imaging conditions, especially iCOM.

This also could lead to what the authors call “residual contrast” (line 236) as simply arising from mistilt in these regions. The 4D STEM data should allow the authors to provide a quantification of how well on-axis the sample was as a function of probe position (again, possibly through the use of simulations), especially in those areas where the “residual contrast” appears, and thus dispel these worries.

3. What do error bars correspond to in the figures (figs. 1bc, 2de, 3e)? It is good to provide error bars, of course, but the authors should clarify how they are calculated.

4. In figure 1, is the zigzag angle absolute (with the reference being the first Nd-Nd interatomic segment measured), or relative (with each angle relative to the previous Nd-Nd atomic pair)? I would suggest to label the zigzag angle plot as “alpha” in the axis title, so the reader can relate the annotation on figure 1a to the plot.

5. What are the horizontal/vertical dimensions in the zigzag angle maps? Are they meant to be a unit-cell-wise maps across

the red boxes (and if not, what are the red boxes)? Or unit-cell-wise across the 36 or 37 positions on the image being analyzed? But then why are they only 5/6 pixels wide? This information is not provided anywhere for the reader. The same comment applies to figure 2c – it's not clear what the physical dimensions of the map are, and/or where each pixel in the figure maps back to on the original iCOM image.

6. I would encourage the authors to merge figures 4 and S6. On its own, it's not clear what the EELS-part of figure 4 shows. The individual F K edges are not particularly instructive, as they look all very similar in both shape and intensity. This is also due to the fact that the spectra have been normalized to arbitrary units: is the normalization (the arbitrary units) the same for apical and basal positions, suggesting almost no variation at all across the field of view of the figure? In turn, Figures 4e-i are almost entirely separate from the EELS, unless the authors want to suggest that the apical or basal sites should have different F K fine structure. These panels would be better separated as their own figure.

7. How should the reader understand figures 3 and 4 together? On the one hand, figure 3 seems to show that there is more F at the center of the nickelate layers. Yet the data in figures 4 and S6 highlight a constant F/O ratio and to the naked eye an almost constant F K intensity (in a.u.) across a distance that is the same size as a full nickelate layer. The authors should clarify this point of the discussion: do they mean to highlight variability across the grown superlattice? And if so, where does each analyzed area belong with respect to the overall film?

8. Generally speaking, for all the various sub-regions analyzed, especially in figures 4 and 5 but perhaps for all high-magnification regions, the authors should offer a larger context image indicating where these sub-regions were cropped from. This is particularly important as for instance the analysis suggests that domains observed in 5a and 5c would correspond to different viewing directions – yet the text suggests that these are found in the same sample within nm distance. Does this suggest the presence of 90 degree unit rotations within the same layer? It would also help the reader relate specific structural distortions with proximity to interfaces, superlattice surface or substrate.

9. The DFT results could be discussed in a bit more detail: they almost appear disconnected from the experimental part of the manuscript. Although the methods section explains that a value of $U=4V$ was chosen, it's not clear how this choice was made (maybe a reference to earlier studies would help validate this choice – although given the very specific structures observed here, the screening will be very specific and so a comment on why a value typical for other nickelates would apply here when both O and F are present).

Similarly to the comment above, I would suggest reminding the reader where the calculated structures appear in the real-life sample (link figures S9a, c, e to experimental images).

10. It is also striking that the higher F intercalation in DFT leads to a higher 2+ character for the Ni – which seems to agree well with the experimental quantification in figure 3e, and yet there is no comment or link to this agreement in the text.

11. There is hardly any discussion of the differences between the two spin channels – yet the conclusion highlights the “high-spin state” (line 300).

12. Small point of detail: how were the maps in figure 3 generated? Simple background subtraction and integration (over what sort of window), or was a model-based approach used?

13. Splitting hair even more: was all the data acquired at 200kV? (it is never explicitly said, but probably needs stating).

Reviewer #2

(Remarks to the Author)

Review on “Atomic-scale observation of geometric frustration in a fluorine-intercalated 1 infinite layer nickelate superlattice” by Chao Yang and coworkers.

Here, C. Yang and collaborators investigate the F-ion intercalation within an infinite layer $NdNiO_{2+x}/SrTiO_3$ superlattice by means of atomic resolution electron microscope and spectroscopy techniques. The incorporation of F into these perovskite-like systems is a challenging task that is worth looking into, since O substitution by F can result in unexpected physical behaviors. For example, the partial substitution of F for O can induce an electronic configuration in the Ni 3d band in the nickelate system, which has been related to the occurrence of superconductivity. The authors combine different advanced electron microscopy techniques to approach the study of the F incorporation into the nickelate later. This is a clever approach, since macroscopically averaged x-ray diffraction techniques might not be of use elucidating these issues, especially in heterostructures where interfaces are not atomically flat (as may be the case here, after F incorporation in the system). These techniques include atomic resolution EELS for chemical imaging and 4D-STEM for atomic column position quantification. EEL spectrum images unambiguously proof the presence of F in the Nd-Ni-O layers, in substitution of O. The F K signal clearly increases while the O signal decreases along the nickelate thickness. These changes also affect the system electronic properties, including the oxidation state of Ni. However, EELS F images do not exhibit a good enough signal-to-noise ratio to ascertain 100% the actual position of F atoms. Here 4D-STEM combined with first principles calculations does the job nicely. Using an iCoM approach, the authors successfully quantify the position of all atomic columns in the system, including the lighter F and O. The incorporation of F affects dramatically the crystal structure of the nickelate, especially the anion octahedral system. Interestingly, the onset of such structural modifications is not atomically sharp, so there is a gradient within the nickelate when approaching the STO interface. Such observation is consistent with

previous findings in perovskite superlattices, and could be due to strains (see for example "Oxygen octahedral distortions in LaMnO₃/SrTiO₃ superlattices". G. Sanchez-Santolino et al., *Microscopy & Microanalysis*, 20, 825-831 (2014).), or perhaps some chemical or electronic disorder ensuing from the growth process (e.g. J. Garcia-Barriocanal et al., *Advanced Materials*, 22, 627-632 (2010).). In any case, the authors convincingly conclude that the NdNiO₂F configuration is the dominant phase within the nickelate layer, with an ordered F configuration in the layers that dramatically affects the system physical properties, rendering a highly insulating state. In this respect, the manuscript is flawless. Being the topic of high interest for the oxide thin film community, as well as for the general materials science audience, I recommend publication as is.

Reviewer #3

(Remarks to the Author)

In "Atomic-scale observation of geometric frustration in a fluorine-intercalated infinite layer nickelate superlattice" Yang et al describe the atomic-level microstructure of NdNiO_xF_y/SrTiO₃ thin film superlattices. Using 4D-STEM and EELS they attempt to identify how the fluorine incorporates into the NdNiO_x lattice and its effects on structural distortions. They argue that in the NdNiO₃ → NdNiO₂ → NdNiO₂F process the fluorine primarily intercalates on the apical oxygen sites though there is substantial mixing on the basal ('equatorial?') sites. This process induces orthorhombic distortions of varying types.

I think the quality and depth of the data and analysis may make this manuscript suitable for eventual publication in *Nat Commun* – certainly the transition metal oxide community will be interested in microstructural analysis of soft chemistry. On the other hand, I found the paper challenging to follow, with large ambiguities in the definition of "domain" and "geometric frustration", and much switching back and forth between effects. The manuscript also does not present basic "under the hood" data that would be helpful for interpretation.

Overall, there are many structural (Nd angles, Ni-O bond angle, F occupation, and all their relative alignments) distinctions and with ambiguous terminology. I would suggest the authors provide either a clear, easy to read table or a multi-paneled crystal structure figure summary with all the structural combinations, clearly labelled as the first or last figure, as a reference? This would help with the readability of the paper.

My biggest confusion is in the terminology "geometric frustration", which the authors do not define. In contemporary times, a casual reader might expect a hexagonal or related lattice to be implicated in this paper. A specialist reader might at least expect a diagram such as that in Fig. 1f, <https://doi.org/10.1038/s41586-022-05681-2>, to clarify what is meant by "geometric frustration". Does "geometric frustration" refer to the competition/coexistence between "domains" A and B in Fig. 5? Given that "geometric frustration" is in the title of the paper, and especially as *Nat Commun* is a broader readership journal, the authors need to obviously and unambiguously illustrate what they mean by Nd geometric frustration.

My other confusion is in the definition of "domain". At first I thought the authors meant the difference between areas with apical and 'basal' occupation of F. Then it seemed like the definition of domain A vs B was just a 90 degree in-plane rotation in the apical-F phase.

1. Can the authors please provide bulk data since STEM is a very local probe? It would be essential to provide thin film XRD and temperature-dependent resistivity data on all of the NdNiO₃/STO, NdNiO₂/STO, and NdNiO_xF_y/STO superlattices shown in this work. Modified nickelates are exquisitely sensitive to parent NdNiO₃ quality so it would be helpful for readers in the community to understand the universality of the observations made in this paper.

2. Can the authors provide more large FOVs in STEM to give a better sense of the global film quality? Also there seems to be substantial amorphisation close to the substrate-film interface (Fig. S2). Do the authors think that comes from reduction or fluorination and why at the interface?

3. Despite the extensive EELS analysis in the manuscript, the authors show very little actual EELS spectra other than in Fig. 4b, c (which I'm not sure what point that is supposed to make). The authors should provide the O K edge spectra used for the analysis in Fig 3c, d; and the Ni L spectra (maybe a waterfall plot) used for the analysis in Fig. 3e.

4. In Fig. 3e the authors use the Ni L₃/L₂ edge intensity ratio to estimate a valence. Can they please provide a reference? I am not aware of this metric being an established proxy for nickel valence.

5. I am confused by the analysis the authors employ to determine F/O occupation. 4D-STEM + DFT to backtrack the F/O occupation seems circuitous.

5a. Have they tried, for example, to look at EELS fine structure and perform a much simpler non-linear least squares fitting at either the O or F (or similar analysis) to spatially map and quantify the O, F distribution? For example similar to this manuscript: <http://dx.doi.org/10.1063/1.4737208>. I would find a direct analysis with EELS more convincing to argue predominant apical F.

6. In the discussion, the authors claim the NdNiO_xF_y samples are not measurable in resistivity due to DFT-predicted bandgaps and Ni²⁺ oxidation state. But Nd₂NiO₄ is measurable (semiconducting). Can the authors show attempted measurements of the resistivity and comment on how sample quality might be related?

7. The group of Steven May (I am not Steven May) has contributed tremendously to studies of fluorination (including studies

of site-occupancy) in transition metal oxide thin films. Though not on nickelates, at least some of their contributions should be properly cited, for example:

7a. <https://doi.org/10.1021/ja410954z>

7b. <https://doi.org/10.1021/acs.chemmater.0c04793>

7c. <https://doi.org/10.1021/acs.inorgchem.0c01148>

8. Have the authors tried fluorinating the perovskite phase? How would that differ from reducing (removing apical oxygen) and then fluorinating? Onozuka (Ref. 1) directly fluorinates without reduction. Is the reduction process and starting from the infinite layer NdNiO₂ state what drives the slight preferential occupation of fluorine of the apical sites? Can either experiments or DFT capture this distinction (between fluorinating NdNiO₃ vs NdNiO₂) or is this just speculation?

9. In Figure S1, the unfluorinated NdNiO₂ looks to already have Nd displacements, i.e. the Nd atoms don't all lie on horizontal planes. Why? All previous examples of NdNiO₂ (Hwang, Ariando) do not show these displacements. Is this from incomplete reduction?

Other minor points:

1. "Basal" oxygen feels misleading, this may imply a square pyramidal configuration. The authors should consider if "equatorial" oxygen would be a more accurate term.
2. Figure 3: Panels c and d represent the intensity profiles extracted from the box in panel a, so it is misleading /hard to read that they are vertically aligned with the entire panel a. Can the authors please properly align just the box in panel a with the intensity line profiles in c and d?
3. Personally I would avoid subjective adjectival phrasing like "definitively demonstrate" or "meticulously quantified".
4. Define " t/λ " in the EELS section, otherwise move this to the methods.
5. Can the authors elaborate on the atom tracking algorithm they use? The positions of the yellow dots look very visually misaligned with the HAADF atoms in Fig. 1...

Version 1:

Reviewer comments:

Reviewer #1

(Remarks to the Author)

In this revised manuscript the authors have successfully clarified all the questions raised, and added important information that strengthens their work. The expanded supplementary file offers very useful additional images and analysis, while the clarification of the EELS data and analysis, now into a simplified Figure 4, vastly improves the flow of the paper.

Similarly, points of technical details have been added to the methods, offering of more complete description of the work. The authors appear to have used the Atomap software package, and mention it in the context of the zigzag angle determination. A reference to the package repository (and access date), alongside with a reference to a paper describing this piece of software, which I believe are not currently included in the reference list, would be welcome.

One final point may need to be briefly addressed (unlikely to require an additional round of reviews). The inclusion of simulated iCOM images is welcome, but it is puzzling to notice a clear mismatch in contrast between the Nd and NiO₂ columns comparing experiments and simulations. In figure 5, in the experimental data, they appear to have almost the same intensity, whereas the iCOM simulations suggest otherwise. I realize these are not meant to lead to quantitative contrast analysis, but the discrepancy is a little troubling - and may point to channeling effects or partial occupancies not fully captured by the models.

A note of explanation on this observation is probably needed before final acceptance of the work - which I am otherwise happy to recommend for publication in Nature Communications.

Reviewer #2

(Remarks to the Author)

Reviewer #3

(Remarks to the Author)

The authors have extensively addressed my comments with excellent thought from the first round of review. The manuscript is well-improved, also especially thanks to the comments from Reviewer 1. I have just a few minor comments left before publication in Nature Communications:

1. The large FOV HAADF-STEM that the authors provided are only shown in the review response but not in the edited manuscript. I would recommend showing these large FOV images in the Supplement as to provide useful sample images to the field. Also, there is significant structural long-range "bending" which looks almost periodic. Do the authors have any thoughts as to why the bending? This looks distinct from the disorder seen in doped NdNiO₂ on SrTiO₃ (no superlattice). Do

the authors think the SrTiO₃ has some effect? I would expect the SrTiO₃ layers to prevent the formation of such bending features.

2. The references on the NiL₃/NiL₂ edge intensity ratio were provided in the response letter but not cited in the updated manuscript.

3. Regarding the fluorinated NdNiO₂ vs NdNiO₃, the authors stated in their first response "However, the significant difference in the degree of fluorination for the two precursor phases makes it difficult to directly compare atomic scale TEM results between the samples."

I would think that larger differences in the bulk-scale x-ray diffraction (which suggest that the NdNiO₂ is more fluorinated) would make it easier to identify what atomic- or meso-scale differences there are in the TEM.

I would hesitate to hold publication for HAADF-STEM on the fluorinated NdNiO₃ samples but I do think such images would be very interesting and useful, even if exquisite atomic-scale analysis is not possible at this stage.

Version 2:

Reviewer comments:

Reviewer #3

(Remarks to the Author)

The authors have fully addressed my remaining points. I recommend for publication and thank them for their complete and high-quality study.

RESPONSE TO REVIEWERS' COMMENTS

Reviewer #1 (Remarks to the Author):

In this manuscript, the authors use detailed atomic-scale scanning transmission electron microscopy imaging and spectroscopy to reveal subtle localized structure inhomogeneities in the F-intercalated layers of a SrTiO₃/infinite-layer nickelate superlattice. 4D-STEM techniques and electron energy loss spectroscopy show how fluorine intercalates at different anion lattice sites, revealing in some localized regions how this leads to unexpected structural distortions. The experimental results are further supported by functional-density-theory-based calculations. This study of oxyfluorides is particularly timely, with numerous research efforts currently directed at exploiting the emerging properties of these intriguing materials, but for which precise control of chemistry and structure is required. The results will thus be of interest to a wide research community beyond expert electron microscopists. The experimental data is excellent and the careful and thorough processing offers unique insights into a system that is set apart by its two-step growth process – some of the O anions being stripped out of the lattice prior to F re-introduction. The authors show convincingly how this leads to unexpected localized geometric distortions of the lattice, with subtle chemical/structural interplay. In turn, these may have important consequences on the properties of the films. Nevertheless, a number of reasonably minor issues should be addressed before this work is accepted for publication, as detailed below.

Response: Thank you very much for your detailed and constructive feedback. We are pleased that you find our study and experimental data valuable and of broad interest to the research community. We have carefully considered the points raised and have addressed all issues in detail in the revised manuscript. Below, we provide a point-by-point response to your comments and outline the changes we have made to improve the clarity and rigor of our study.

1. The authors rely on the iCOM technique to study the structural displacement of light-element atomic columns (O and F). The observation of several distinct structures in separate domains and from the same images/datasets certainly lends credence to the interpretation in terms of octahedral tilts and distortions depending on the presence of O or F at apical or basal sites. Nevertheless, as with all phase-contrast-based imaging (in a broad sense – iCOM is of course very specific), channeling and other phase effects may affect the image formation, especially in the thicker areas of the sample and given the presence of heavy cation columns such as Nd. Given that the authors have DFT-optimized models, simulations of expected iCOM contrast would offer a further welcome validation of the image interpretation, and they should consider adding these simulations to a revised manuscript.

Response: Thank you for the constructive and insightful comments of the reviewer. We have added the iCoM image simulations based on the relaxed structures derived from the DFT calculations, which are in close agreement with the experimental results. We have modified Figure 5 below and corrected the corresponding text in the manuscript. In addition, we have added another simulated image as shown in Figure R2 (Figure S10 in the manuscript) to show the difference in Nd distortion due to F intercalation.

Figure R1: Comparison between the experimentally observed structure distortion and the relaxed structure model from DFT calculations. (a) Reconstructed iCoM image of domain A with clear polyhedral rotation and (b) the relaxed structure model with F intercalation at apical sites. (c) A simulated iCoM image of the structure in (b). The red dashed lines mark the anion sublattice distortion. (d) Reconstructed iCoM image of domain B showing a clear zigzag arrangement of Nd atoms and (e) the relaxed structure model with F intercalation at apical sites. (f) A simulated iCoM image of the structure in (e). The blue arrows indicate the displacement direction of the Nd atoms. The red dashed line marks the anion sublattice without distortion. Domain B is formed by a 90° rotation of domain A. (g) A relaxed structural model with two connected domains. (h) Reconstructed iCoM image of the domain interface and (i) the enlarged structural model. (j) A simulated iCoM image of the structure in (i). (k) Reconstructed iCoM image of the region with abnormal structure distortion and (l) the structure model from the DFT calculations with partial intercalation of F-ions at the basal sites. (m) A simulated iCoM image of the structure in (l).

Figure R2: (a) HAADF image of the region with elliptical Nd columns pointing in the same direction. (b) The corresponding structure model with F intercalation at apical sites for viewing direction [100] and (c) the simulated iCoM image. (d) HAADF image of the region with perpendicular elliptical Nd columns. (e) The corresponding structure model with F intercalation at apical sites for viewing direction [100] and (f) the simulated iCoM image.

columns. (e) The corresponding structure models with F intercalation at basal sites for the viewing direction [100] and (f) the simulated iCoM image.

2. One additional worry is local tilt along the beam direction. Figure S2 suggests the domains are separated by a few to 10s of nm, a distance over which the local crystal tilt could possibly change enough to affect imaging conditions, especially iCOM. This also could lead to what the authors call “residual contrast” (line 236) as simply arising from mistilt in these regions. The 4D STEM data should allow the authors to provide a quantification of how well on-axis the sample was as a function of probe position (again, possibly through the use of simulations), especially in those areas where the “residual contrast” appears, and thus dispel these worries.

Response: Thank you for the reviewer’s insightful comments. As shown in Figure R1e, and f, the observed contrast is attributed to the anions at the basal sites in close proximity to the Nd atoms, which is a consequence of the distortion that occurs after F intercalation, despite the absence of any tilt. In addition, Figure R7 shows the large region for the cropped iCoM images in Figure 5, where the SrTiO₃ layer serves as a reference for the absence of any tilting. We have corrected for “residual contrast” because it may be an inaccurate description, as the contrast should be present in a normal structure. In contrast, the contrast is absent in the case of a structural model shown in Figure R11, and m.

3. What do error bars correspond to in the figures (figs. 1bc, 2de, 3e)? It is good to provide error bars, of course, but the authors should clarify how they are calculated.

Response: Thank you for the reviewer’s insightful and rigorous suggestion. The description of the error bars has been added to the figure captions.

4. In figure 1, is the zigzag angle absolute (with the reference being the first Nd-Nd interatomic segment measured), or relative (with each angle relative to the previous Nd-Nd atomic pair)? I would suggest to label the zigzag angle plot as “alpha” in the axis title, so the reader can relate the annotation on figure 1a to the plot.

Response: We appreciate the reviewer’s comments. Since there is a reference value of the Sr-Sr angle in the SrTiO₃ layer, Figure 1 shows the absolute values of the zigzag angles. Thank you for the reviewer’s rigorous suggestion. We have changed the designation of the *x* axis title to “alpha” and corrected the descriptions in the figure caption.

5. What are the horizontal/vertical dimensions in the zigzag angle maps? Are they meant to be a unit-cell-wise maps across the red boxes (and if not, what are the red boxes)? Or unit-cell-wise across the 36 or 37 positions on the image being analyzed? But then why are they only 5/6 pixels wide? This information is not provided anywhere for the reader. The same comment applies to figure 2c – it’s not clear what the physical dimensions of the map are, and/or where each pixel in the figure maps back to on the original iCOM image.

Response: We appreciate the reviewer’s important question. We apologize for the misunderstanding of the results. The horizontal and vertical axes indicate the positions and corresponding zigzag angle values, respectively. In addition, the following figures clarify the exact regions used for the zigzag angle calculations. In Figure R3, thirteen Nd atom points were used to calculate 6 zigzag angles per row, resulting in a 6 × 36 map in Figure 1 b and c. In Figure R4, seven Nd atom points were used to calculate 3 zigzag angles per row, resulting in a 3 × 25 map in Figure 2c. We have added the following figures to the Supplementary Information and added the corresponding information to the captions of Figure 1 and 2.

Figure R3: (a) HAADF image of the $8\text{NdNiO}_2/4\text{SrTiO}_3$ superlattice sample, together with an overlay plot of the marked Nd columns. (b) HAADF image of the $8\text{NdNiO}_2\text{F}_x/4\text{SrTiO}_3$ superlattice sample, together with an overlay plot of the marked Nd columns.

Figure R4: HAADF image of the $8\text{NdNiO}_2\text{F}_x/2\text{SrTiO}_3$ superlattice sample, together with an overlay plot of the marked Nd columns.

6. I would encourage the authors to merge figures 4 and S6. On its own, it's not clear what the EELS-part of figure 4 shows. The individual F K edges are not particularly instructive, as they look all very similar in both shape and intensity. This is also due to the fact that the spectra have been normalized to arbitrary units: is the normalization (the arbitrary units) the same for apical and basal positions,

suggesting almost no variation at all across the field of view of the figure? In turn, Figures 4e-i are almost entirely separate from the EELS, unless the authors want to suggest that the apical or basal sites should have different F K fine structure. These panels would be better separated as their own figure.

Response: We thank the reviewer for the constructive suggestions. We extracted the F signal at the same integrated spot region without normalization in order to check the relative intensity variation, as normalization may change the relative F intensity for different positions. Since there is almost no variation in the F signal in the NdNiO_2F_x inner layer and so we calculated the O/F relative signal intensity ratio at both apical and basal locations to understand the possible occupancy of F at both locations. We have modified the Figure 4 below (Figure R5). We have also included Figure 4d-i in Figure S6 (Figure R2).

Figure R5: Detected F-signal at both basal and apical sites and its effect on the atomic structure distortion. (a) Overview and (b) enlarged HAADF images of the region for EELS measurements of the NdNiO_2F_x inner layer. Red and yellow spots mark the apical and basal sites, respectively. (c) The F signal extracted from the red spots in (b). (d) The F signal extracted from the yellow spots in (b). (e) The relative O/F signal intensity ratio at the apical and basal sites.

7. How should the reader understand figures 3 and 4 together? On the one hand, figure 3 seems to show that there is more F at the center of the nickelate layers. Yet the data in figures 4 and S6 highlight a constant F/O ratio and to the naked eye an almost constant F K intensity (in a.u.) across a distance that is the same size as a full nickelate layer. The authors should clarify this point of the discussion: do they mean to highlight variability across the grown superlattice? And if so, where does each analyzed area belong with respect to the overall film?

Response: We appreciate the reviewer's comments. In Figure 3, it is clear that less F signal is observed only at the interfaces. However, the signal intensity in the nickelate inner layer is almost maintained and the corresponding Ni valence is also about 2+. The dashed blue lines in Figure 3 are not fitting lines, but only to show the variations. They may show an erroneous indication of more F in the center of the nickelate layers, which has been corrected in the Figure R6 below.

In Figures 4 and S6, the signal intensity was extracted from specific positions in the inner nickelate layer (see Figure R5) to analyze the F signals at apical and basal positions, which cannot be deduced from Figure 3. Figure 4 provides evidence from EELS results and atomic structures showing how F intercalates at the basal sites. For this reason, we have included the images in Figure 4d-i along with

the EELS results. Following the reviewer's 6th comment, we have removed Figure 4d-i and included it in Figure S6, which contains more simulation results.

Figure R6: Elemental distribution, composition, and valence state in the oxyfluoride NdNiO₂F superlattice. (a) ADF image for the EELS measurement of the 8NdNiO₂F/4SrTiO₃ superlattice. (b) The corresponding elemental distributions: Nd (purple), Ni (yellow), Sr (blue), Ti (green), O (red), and F (orange). (c) Intensity line profile of the O K edge signal extracted from the region marked by the yellow dashed box in (a). (d) Intensity line profile of the F K edge signal extracted from the region marked by the red dashed box in (a). (e) The calculated Ni L₃/L₂ intensity ratio in the nickelate layer. The dashed lines indicate the reference values of the Ni L₃/L₂ intensity ratio for different Ni valence states, which are derived from NdNiO₃ for Ni³⁺,⁴¹ NiO for Ni²⁺,⁴² and NdNiO₂ for Ni⁺.¹³ The error bars in panel (e) were calculated from multiple data sets.

8. Generally speaking, for all the various sub-regions analyzed, especially in figures 4 and 5 but perhaps for all high-magnification regions, the authors should offer a larger context image indicating where these sub-regions were cropped from. This is particularly important as for instance the analysis suggests that domains observed in 5a and 5c would correspond to different viewing directions – yet the text suggests that these are found in the same sample within nm distance. Does this suggest the presence of 90 degree unit rotations within the same layer? It would also help the reader relate specific structural distortions with proximity to interfaces, superlattice surface or substrate.

Response: We appreciate the reviewer's constructive comments and have made the following changes to the figures as suggested. The regions where the subregions have been cropped are now shown: Figure R7 corresponds to Figure 5, and Figure R8 corresponds to Figure 4. Our results show 90-degree unit rotations within the same layer and their interconnection across multiple unit cells, forming domains. These details have been added to the Supplementary Information (Figures S9, S12) due to the extensive content in the main figures.

Figure R7: (a) Overview HAADF image of the $\text{NdNiO}_2\text{F}_x/\text{SrTiO}_3$ superlattice sample. The yellow dashed box marks the region for the 4D-STEM measurement. Reconstructed (b) ADF and (c) iCoM images for the enlarged region. The yellow dashed boxes in (c) show the regions of (a, c, f, and h) in Figure 5 in the main text.

Figure R8: (a) Overview and (b) enlarged HAADF images of the $\text{NdNiO}_2\text{F}_x/\text{SrTiO}_3$ superlattice sample. (c) HAADF image of the region with elliptical Nd columns pointing in the same direction. The corresponding structure models with F intercalation at apical sites for viewing directions $[100]$ in (d) and $[010]$ in (e), respectively. (f) HAADF image of the region with perpendicular elliptical Nd columns. The corresponding structure models with F intercalation at basal sites for the viewing directions (g) $[100]$ and (h) $[010]$, respectively. The red boxes labeled c, c', f, and f' in (b) indicate the analogous enlarged regions of (c) and (f).

9. The DFT results could be discussed in a bit more detail: they almost appear disconnected from the experimental part of the manuscript. Although the methods section explains that a value of $U=4\text{V}$ was chosen, it's not clear how this choice was made (maybe a reference to earlier studies would help validate

this choice – although given the very specific structures observed here, the screening will be very specific and so a comment on why a value typical for other nickelates would apply here when both O and F are present). Similarly to the comment above, I would suggest reminding the reader where the calculated structures appear in the real-life sample (link figures S9a, c, e to experimental images).

Response: We appreciate the reviewer's comments. We have mainly compared the atomic structures between the DFT calculation and the experimental results, which are in agreement with each other. We have added more simulations according to the DFT results as suggested. We have added a little more discussion about the density of state results from DFT on page 8. Regarding the choice of U value, we have considered the reference papers of related nickelates. The agreement of the band gap values between the DFT calculation and the experimental result is an important factor in the choice of the U value, although both O and F are present. The U value of 4V is still available in our case. In addition, we have added the description in the figure caption to link Figures S9a, c, e to the experimental images in Figures 5d, S9f and Figure 5k.

10. It is also striking that the higher F intercalation in DFT leads to a higher 2+ character for the Ni – which seems to agree well with the experimental quantification in figure 3e, and yet there is no comment or link to this agreement in the text.

Response: We appreciate the reviewer's constructive comments. We have changed the text on page 8 to link the EELS result to the DFT calculations.

11. There is hardly any discussion of the differences between the two spin channels – yet the conclusion highlights the “high-spin state” (line 300).

Response: We appreciate the reviewer's strong comments. We have corrected and removed the description of the “high spin state”.

12. Small point of detail: how were the maps in figure 3 generated? Simple background subtraction and integration (over what sort of window), or was a model-based approach used?

Response: We appreciate the reviewer's questions. The EELS map was generated using the EELS quantification module of the Digitalmicrograph software. The background model is the standard power-law model. The cross section model is the Hartree-Slater model. The EELS data were denoised using the PCA method to obtain the EELS maps. We have added more details in the experimental part.

13. Splitting hair even more: was all the data acquired at 200kV? (it is never explicitly said, but probably needs stating).

Response: We appreciate the reviewer's rigorous comments. All STEM results were obtained at 200 kV, and we have added the description in the experimental section.

Reviewer #2 (Remarks to the Author):

Review on “Atomic-scale observation of geometric frustration in a fluorine-intercalated 1 infinite layer nickelate superlattice” by Chao Yang and coworkers.

Here, C. Yang and collaborators investigate the F-ion intercalation within an infinite layer $\text{NdNiO}_{2+x}/\text{SrTiO}_3$ superlattice by means of atomic resolution electron microscope and spectroscopy techniques. The incorporation of F into these perovskite-like systems is a challenging task that is worth looking into, since O substitution by F can result in unexpected physical behaviors. For example, the partial substitution of F for O can induce an electronic configuration in the Ni 3d band in the nickelate system, which has been related to the occurrence of superconductivity. The authors combine different advanced electron microscopy techniques to approach the study of the F incorporation into the nickelate later. This is a clever approach, since macroscopically averaged x-ray diffraction techniques might not be of use elucidating these issues, especially in heterostructures where interfaces are not atomically flat (as may be the case here, after F incorporation in the system). These techniques include atomic resolution EELS for chemical imaging and 4D-STEM for atomic column position quantification. EEL spectrum images unambiguously prove the presence of F in the Nd-Ni-O layers, in substitution of O. The F K signal clearly increases while the O signal decreases along the nickelate thickness. These changes also affect the system electronic properties, including the oxidation state of Ni. However, EELS F images do not exhibit a good enough signal-to-noise ratio to ascertain 100% the actual position of F atoms. Here 4D-STEM combined with first principles calculations does the job nicely. Using an iCoM approach, the authors successfully quantify the position of all atomic columns in the system, including the lighter F and O. The incorporation of F affects dramatically the crystal structure of the nickelate, especially the anion octahedral system. Interestingly, the onset of such structural modifications is not atomically sharp, so there is a gradient within the nickelate when approaching the STO interface. Such observation is consistent with previous findings in perovskite superlattices, and could be due to strains (see for example “Oxygen octahedral distortions in $\text{LaMnO}_3/\text{SrTiO}_3$ superlattices”. G. Sanchez-Santolino et al., *Microscopy & Microanalysis*, 20, 825-831 (2014).), or perhaps some chemical or electronic disorder ensuing from the growth process (e.g. J. Garcia-Barriocanal et al., *Advanced Materials*, 22, 627-632 (2010).). In any case, the authors convincingly conclude that the NdNiO_2F configuration is the dominant phase within the nickelate layer, with an ordered F configuration in the layers that dramatically affects the system physical properties, rendering a highly insulating state. In this respect, the manuscript is flawless. Being the topic of high interest for the oxide thin film community, as well as for the general materials science audience, I recommend publication as is.

Response: We appreciate the reviewer's rigorous comments. We sincerely thank the reviewer for the positive and detailed evaluation of our work. We appreciate the recognition of the challenges associated with F-ion intercalation in nickelates and for acknowledging the importance of our study to both the oxide thin film community and the broader materials science audience. We are pleased that the reviewer finds our combination of advanced microscopy techniques, including atomic-resolution EELS and 4D-STEM, along with first-principles calculations, to be an effective approach for investigating F-incorporation and its effects on the nickelate structure and electronic properties. The reviewer's observations regarding the structural gradient near the STO interface are particularly insightful, and we appreciate the references to related work that agree well with our findings. We have added these related references to our manuscript. These comparisons further highlight the broader relevance of our findings. We are very pleased that the reviewer finds the manuscript suitable for publication as is, and we greatly appreciate his/her support and encouraging feedback. Thank you again for your thoughtful review.

Reviewer #3 (Remarks to the Author):

In "Atomic-scale observation of geometric frustration in a fluorine-intercalated infinite layer nickelate superlattice" Yang et al describe the atomic-level microstructure of NdNiO_xF_y/SrTiO₃ thin film superlattices. Using 4D-STEM and EELS they attempt to identify how the fluorine incorporates into the NdNiO_x lattice and its effects on structural distortions. They argue that in the NdNiO₃ → NdNiO₂ → NdNiO₂F process the fluorine primarily intercalates on the apical oxygen sites though there is substantial mixing on the basal ('equatorial') sites. This process induces orthorhombic distortions of varying types. I think the quality and depth of the data and analysis may make this manuscript suitable for eventual publication in Nat Commun – certainly the transition metal oxide community will be interested in microstructural analysis of soft chemistry. On the other hand, I found the paper challenging to follow, with large ambiguities in the definition of "domain" and "geometric frustration", and much switching back and forth between effects. The manuscript also does not present basic "under the hood" data that would be helpful for interpretation.

Response: We would like to thank the reviewer for his/her thoughtful and constructive feedback on our manuscript. We appreciate the positive assessment of the quality and potential of our data and analysis. To improve clarity, we have revised the manuscript to provide more precise descriptions of the "domain" in the main text when the term is mentioned. The term "domain" has been defined at the beginning of the main text, where it was first mentioned on page 4. In addition, we have corrected the "geometric frustration" to "geometric reconstruction" to avoid any confusion since domains are the key words in the manuscript. Geometric reconstruction is available to describe the atomic and structural rearrangements at the domain interface, which is a more intuitive and understandable term.

Overall, there are many structural (Nd angles, Ni-O bond angle, F occupation, and all their relative alignments) distinctions and with ambiguous terminology. I would suggest the authors provide either a clear, easy to read table or a multi-paneled crystal structure figure summary with all the structural combinations, clearly labelled as the first or last figure, as a reference? This would help with the readability of the paper.

Response: We would like to thank the reviewer for the constructive insights. We have modified Figure 1 below to highlight all relevant parameters together at the beginning, including the zigzag angle (α) of the Nd atoms, the Ni-O-Ni angle (β), and the F/O occupation, which we hope will provide readers with a clearer understanding. In addition, we have also added the overview HAADF image to show the region where we have cropped the locally enlarged HAADF and iCoM images in Figure 5, as shown in Figure R8.

Figure R9: Synthesis of the fluorine-intercalated infinite layer nickelates and the formation of different domains in the oxyfluoride NdNiO_2F . (a) Structural models of the perovskite phase, the infinite layer phase and the oxyfluoride phase. α denotes the zigzag angle α between the A-site cations, specifically the Nd and Sr ions. β is for the Ni-O-Ni angle. The red and gray dashed circles indicate the potential for occupancy of the O and F atoms at the basal and apical sites, respectively. (b) HAADF image of the $8\text{NdNiO}_2/4\text{SrTiO}_3$ superlattice, a map of the zigzag angle α , and a line profile of the zigzag angle α between the A-site cations (A: Nd and Sr). (c) HAADF image of the $8\text{NdNiO}_2\text{F}/4\text{SrTiO}_3$ superlattice, a map of the zigzag angle α , and a line profile of the zigzag angle α between the A-site cations (A: Nd and Sr). Thirteen Nd atom points were used to calculate 6 zigzag angles per row, resulting in a 6×36 map in (b) and (c). The exact regions used to calculate the zigzag angles in (b) and (c) are shown in Figure S2. The error bars in panels (b) and (c) are derived from the Gaussian fit using the Atomap package. (d) A magnified HAADF image of the region marked by the red dashed box in (b). (e) A magnified HAADF image of the region marked by the red dashed box in (c) showing two domains. In (d) and (e), the yellow dots and the red and blue lines are guides for the eye, indicating the positions of the A-site cations and the straight and zigzag arrangements of the cations, respectively.

My biggest confusion is in the terminology "geometric frustration", which the authors do not define. In contemporary times, a casual reader might expect a hexagonal or related lattice to be implicated in this paper. A specialist reader might at least expect a diagram such as that in Fig. 1f, <https://doi.org/10.1038/s41586-022-05681-2>, to clarify what is meant by "geometric frustration". Does

"geometric frustration" refer to the competition/coexistence between "domains" A and B in Fig. 5? Given that "geometric frustration" is in the title of the paper, and especially as Nat Commun is a broader readership journal, the authors need to obviously and unambiguously illustrate what they mean by Nd geometric frustration.

Response: Thanks for your insightful comment. We apologize for the confusion regarding the use of "geometric frustration." After reviewing the terminology, we have replaced "geometric frustration" with "geometric reconstruction" in the revised manuscript. "Geometric reconstruction" more accurately describes the structural rearrangements that occur at the interface between domains A and B, as shown in Figure 5. This term specifically refers to the atomic and bond adjustments made to accommodate the mismatch between the two domains. In contrast, "geometric frustration" has been well used in hexagonal or related lattices in the spin systems and can lead to potential confusion, although it can also arise from competing distortions of the oxygen octahedra or lattice rotations in complex oxide materials. We have updated the manuscript accordingly, and we hope that this clarification resolves the confusion for both general and specialized readers. Figure 5g may help visualize the meaning of "geometric reconstruction" at the domain interface.

My other confusion is in the definition of "domain". At first I thought the authors meant the difference between areas with apical and 'basal' occupation of F. Then it seemed like the definition of domain A vs B was just a 90 degree in-plane rotation in the apical-F phase.

Response: We would like to thank the reviewer for this comment. We have highlighted the definition of "domain" in the manuscript to avoid the confusion. The term "domain" has been defined at the beginning of the main text, where it is first mentioned on page 4.

1. Can the authors please provide bulk data since STEM is a very local probe? It would be essential to provide thin film XRD and temperature-dependent resistivity data on all of the NdNiO₃/STO, NdNiO₂/STO, and NdNiO_xF_y/STO superlattices shown in this work. Modified nickelates are exquisitely sensitive to parent NdNiO₃ quality so it would be helpful for readers in the community to understand the universality of the observations made in this paper.

Response: We appreciate the thoughtful comments of the reviewer. The XRD results for the perovskite nickelate/SrTiO₃ superlattice, the infinite layer nickelate/SrTiO₃ superlattice, the fluorine-doped perovskite nickelate/SrTiO₃ superlattice, and the fluorine-doped infinite layer nickelate/SrTiO₃ superlattice have been included in the revised manuscript. There is a clear difference in the fluorination process for these two precursors. Specifically, it took 168 hours to fluorinate the perovskite nickelate/SrTiO₃ superlattice at 350°C, while only 24 hours were required to fluorinate the infinite-layer nickelate/SrTiO₃ superlattice. The former was not fully fluorinated, as evidenced by the NNOF peak still being visible in the XRD results, although it is significantly diminished.

The NNOF peaks in the XRD results are consistent with the results reported by Onozuka (Ref. 1). Furthermore, the temperature-dependent resistivity of the pristine and reduced superlattice samples has been discussed in another published literature (Ref. <http://dx.doi.org/10.18419/opus-12690>), and related TEM results are available in our previous work (Ref. Nano Lett. 2023, 23, 8, 3291–3297). Regarding the temperature-dependent resistivity of the fluorine-doped samples, we regret that we were unable to measure it with our current experimental setup (used to measure the electrical properties from room temperature to low temperature) due to the high insulating nature of the samples, which is also reported in Ref.1 and another similar system (Ref. APL Mater. 8, 091112 (2020)).

Figure R10: XRD results of the perovskite nickelate/SrTiO₃ superlattice, the infinite layer nickelate/SrTiO₃ superlattice, the fluorine-doped perovskite nickelate/SrTiO₃ superlattice, and the fluorine-doped infinite layer nickelate/SrTiO₃ superlattice samples.

2. Can the authors provide more large FOVs in STEM to give a better sense of the global film quality? Also there seems to be substantial amorphisation close to the substrate-film interface (Fig. S2). Do the authors think that comes from reduction or fluorination and why at the interface?

Response: We appreciate the reviewer's comments. We have included an ADF image with a larger field of view (FOV) at low resolution, where the disordered phases are still visible in the first NdNiO₂ or NdNiO₂F_x layer. The disordered phases were formed during the reduction process, which may be influenced by the strain from the substrate.

Figure R11: Overview HAADF images of (a) $8\text{NdNiO}_2\text{F}_x/4\text{SrTiO}_3$ superlattice and (b) $8\text{NdNiO}_2/4\text{SrTiO}_3$ superlattice.

3. Despite the extensive EELS analysis in the manuscript, the authors show very little actual EELS spectra other than in Fig. 4b, c (which I'm not sure what point that is supposed to make). The authors should provide the O K edge spectra used for the analysis in Fig 3c, d; and the Ni L spectra (maybe a waterfall plot) used for the analysis in Fig. 3e.

Response: We appreciate the reviewer's comments. To better illustrate the variation in Ni valence, we have plotted the white-line ratio of the Ni L-edges in Figure 3e. Following the reviewer's rigorous suggestion, we have added the Ni L-edges and O K-edges to the Supplementary Information (Figure S6), as shown in the figure below.

Figure R12: (a) HAADF image of the region for EELS measurements. EELS spectra of (b) O K edges and (c) Ni L edges extracted from the region marked with yellow dashed boxes in (a).

4. In Fig. 3e the authors use the Ni L3/L2 edge intensity ratio to estimate a valence. Can they please provide a reference? I am not aware of this metric being an established proxy for nickel valence.

Response: We appreciate the insightful comments of the reviewer. The white-line ratio of the Ni-L edges can be used to estimate the valence of Ni, and this approach is well documented in the literature. Further references can be found in the following sources: Appl. Phys. Lett. 53, 1405–1407 (1988); Phys. Rev. B 69, 235103 (2004); Microsc. Microanal. 16, 1458–1459 (2010); APL Mater. 4, 096105 (2016); Adv. Electron. Mater. 3, 1700321 (2017); Scripta Materialia 197, 113790 (2021).

5. I am confused by the analysis the authors employ to determine F/O occupation. 4D-STEM + DFT to backtrack the F/O occupation seems circuitous.

Response: We appreciate the insightful comments of the reviewer. Using the iCoM method in 4D-STEM, we were able to obtain a high-resolution image of the F and O sublattices. However, distinguishing between the F and O sites remains a challenge. By combining this approach with DFT calculations, we can compare different configurations of F and O to identify the most reliable F ion intercalation based on the associated atomic structure distortions.

5a. Have they tried, for example, to look at EELS fine structure and perform a much simpler non-linear least squares fitting at either the O or F (or similar analysis) to spatially map and quantify the O, F distribution? For example similar to this manuscript: <http://dx.doi.org/10.1063/1.4737208>. I would find a direct analysis with EELS more convincing to argue predominant apical F.

Response: We sincerely appreciate the reviewer's professional suggestion. As shown below, the fitted F signal map is noisy due to the low F signal intensity when using the non-linear least-squares fit, similar to the map shown in Figure b. This makes it difficult to obtain an atomic-scale elemental map of the F distribution. This limitation further emphasizes the importance of combining 4D-STEM, DFT, and EELS techniques to reliably identify F intercalation.

Figure R13: HAADF image of the region for EELS measurement and the EELS map of the F elements, obtained using non-linear least squares (NLLS) fitting in Digitalmicrograph.

6. In the discussion, the authors claim the NdNiO_xF_y samples are not measurable in resistivity due to DFT-predicted bandgaps and Ni^{2+} oxidation state. But Nd_2NiO_4 is measurable (semiconducting). Can the authors show attempted measurements of the resistivity and comment on how sample quality might be related?

Response: We appreciate the insightful comments of the reviewer. The NdNiO_2F_x film, synthesized by direct fluorination of the NdNiO_3 film, exhibits a highly insulating state, with resistivity exceeding $10^4 \Omega \cdot \text{cm}$, as reported in ref. 1 (ACS Appl. Mater. Interfaces, 9, 12, 10882, 2017). This insulating state is also outside of the measurement limit. A distinct color transition is observed from metallic NdNiO_3 (black) to $\text{NdNiO}_{3-x}\text{F}_x$ (transparent) is observed at room temperature (ref. 1), which serves as a simple visual indicator of its insulating nature. This color transition is also observed in our sample. Since the experimental setup in our laboratory is used to measure the electrical properties from room temperature to low temperature, and the resistance values are higher than the upper limit of our measurement device i.e. PPMS., this is the reason for the statement in the manuscript that “resistivity measurements of the fluorinated superlattice show highly insulating behavior, so that it was not possible to measure its temperature dependence.” in the manuscript. In addition, we do not know what interesting low temperature physics in the electrical properties can be studied for this highly insulating sample. The resistance switching from metal to insulating may be more interesting in the related systems (Ref. 1). Or the synthesis of the superconducting phase in F-doped infinite layer NdNiO_2 is another interesting topic. These are exciting directions for future experimental work. Furthermore, DFT calculations confirm that F intercalation opens a band gap, in agreement with the results in Refs. 1 and 7.

In the case of the 214 phase of nickelates, we found the insulating behavior of Nd_2NiO_4 (ref. Phys. Rev. B 104, 184518 (2020)) within a short temperature range of 150 K to 200 K, where it may be outside the measurement limit at temperatures below 150 K. Similarly, we found a highly insulating La_2NiO_4 sample, where it is also outside of the measurement limit. (Ref. APL Mater. 8, 091112 (2020)). In addition, we found that the $\text{Re}_2\text{NiO}_{4+\delta}$ (Re: La, Nd, or Pr) is semiconducting or conducting, which is related to the ionic hopping between interstitial oxygen sites. (Ref., J. Am. Chem. Soc. 130, 9, 2762 (2008); J. Mater. Chem., 16, 3402 (2006); Solid State Ion., 176, 2717 (2005))

7. The group of Steven May (I am not Steven May) has contributed tremendously to studies of fluorination (including studies of site-occupancy) in transition metal oxide thin films. Though not on nickelates, at least some of their contributions should be properly cited, for example:

7a. <https://doi.org/10.1021/ja410954z>

7b. <https://doi.org/10.1021/acs.chemmater.0c04793>

7c. <https://doi.org/10.1021/acs.inorgchem.0c01148>

Response: We appreciate the reviewer's suggestion. We have added the related references in our manuscript.

8. Have the authors tried fluorinating the perovskite phase? How would that differ from reducing (removing apical oxygen) and then fluorinating? Onozuka (Ref. 1) directly fluorinates without reduction. Is the reduction process and starting from the infinite layer NdNiO_2 state what drives the slight preferential occupation of fluorine of the apical sites? Can either experiments or DFT capture this distinction (between fluorinating NdNiO_3 vs NdNiO_2) or is this just speculation?

Response: We thank the reviewer for their insightful question regarding the potential differences between directly fluorinating the perovskite NdNiO_3 phase versus reducing it to the infinite-layer NdNiO_2 phase prior to fluorination. Our study focuses specifically on fluorinating the infinite-layer NdNiO_2 phase as a starting material, motivated by its unique structural properties and potential for tunable electronic properties upon fluorination. The choice of NdNiO_2 as a precursor was driven by its established synthesis methods and its relevance in the context of layered nickelates with emergent electronic behaviors.

The reviewer raises an important question about the potential role of the reduction process in influencing fluorine site occupancy. While we do not currently have experimental data for the direct fluorination of NdNiO_3 , it is plausible that the reduction to NdNiO_2 , which removes the apical oxygen, creates an environment that slightly favors fluorine occupancy at the apical sites. This hypothesis is consistent with the structural and electronic differences between the two phases: NdNiO_3 retains its octahedral coordination around Ni, while NdNiO_2 adopts a square-planar geometry with more accessible apical positions. Direct fluorination of NdNiO_3 would likely lead to distinct site preferences, as supported by theoretical calculations (Ref. 4 in the manuscript: J. Phys. Chem. C, 123, 51, 31190 (2019)).

We agree that a systematic comparison between the fluorination of NdNiO_3 and NdNiO_2 would provide valuable insights into the role of the precursor phase and the reduction process. We performed a direct fluorination of the as-grown $\text{NdNiO}_3/\text{SrTiO}_3$ superlattices without going through the intermediate $\text{NdNiO}_2/\text{SrTiO}_3$ reduced state. The time required to fluorinate samples was significantly larger in the perovskite phase than in the infinite-layer phase i.e a minimum of one week for $\text{NdNiO}_3/\text{SrTiO}_3$ and times as low as one hour in the $\text{NdNiO}_2/\text{SrTiO}_3$ samples. From the X-ray diffraction measurements shown in Fig. R10, one can see that the sample with main Bragg peak at lower angles is the one which went through the intermediate reduced state. The Bragg peak position gives a rough estimation of the degree of fluorination of the samples [Onozuka et. al.] However, the significant difference in the degree of fluorination for the two precursor phases makes it difficult to directly compare atomic scale TEM results between the samples.

While our current TEM study does not address the fluorination of NdNiO₃, we believe that the findings on fluorinated NdNiO₂ provide an important foundation for understanding fluorine site preferences in nickelates. We appreciate the reviewer's suggestion and see it as an exciting direction for future experimental and computational investigations to further explore the differences between these phases.

9. In Figure S1, the unfluorinated NdNiO₂ looks to already have Nd displacements, i.e. the Nd atoms don't all lie on horizontal planes. Why? All previous examples of NdNiO₂ (Hwang, Ariando) do not show these displacements. Is this from incomplete reduction?

Response: We thank the reviewer for pointing this out. In fact, there should be no Nd displacements in the non-fluorinated NdNiO₂, as shown in Figure 1b of our manuscript. The Nd displacements observed in Figure S1 are due to sample drift, which is also evident in the SrTiO₃ layer. We regret any confusion caused by this oversight and have corrected Figure S1 accordingly, using the SrTiO₃ layer as the reference for the drift correction. The revised figure is shown below.

Figure R14: Reconstructed (a) ADF and (b) iCoM images of the NdNiO₂/SrTiO₃ superlattice sample. The infinite layer structure model is shown on the right side.

Other minor points:

1. "Basal" oxygen feels misleading, this may imply a square pyramidal configuration. The authors should consider if "equatorial" oxygen would be a more accurate term.

Response: We appreciate the reviewer's comment. We have carefully reviewed the published work in the related systems, where the basal oxygen is often used in octahedral coordination. However, we believe that the reviewer's comment is a valuable one and have highlighted the "equatorial" oxygen in

the manuscript where the basal oxygen is first mentioned, which provides a more accurate representation.

2. Figure 3: Panels c and d represent the intensity profiles extracted from the box in panel a, so it is misleading /hard to read that they are vertically aligned with the entire panel a. Can the authors please properly align just the box in panel a with the intensity line profiles in c and d?

Response: We appreciate the reviewer's suggestion. We have modified the figure and added additional markers to avoid misleading information. The revised figure is shown in Figure R6.

3. Personally I would avoid subjective adjectival phrasing like “definitively demonstrate” or “meticulously quantified”.

Response: We appreciate the reviewer's comments. We have removed these descriptions from the manuscript.

4. Define “ t/λ ” in the EELS section, otherwise move this to the methods.

Response: We appreciate the reviewer's comments. We have added the description in the Methods section.

5. Can the authors elaborate on the atom tracking algorithm they use? The positions of the yellow dots look very visually misaligned with the HAADF atoms in Fig. 1...

Response: We appreciate the reviewer's comments. The atom tracking algorithms used in Figures 1b and 1c are the Gaussian function and the center of mass function for atom refinements in the Python library of Atomap. The Supplementary Information has been updated to include Figures R3 and R4, which provide a detailed illustration of the fitting region. As shown in Figure R3, the red dots are from the fit. In Figure 1d, the dots and lines are guides for the eye showing straight and zigzag cation arrangements that are not from fitting. We apologize for this misunderstanding and have corrected the figure (Figure R9) and the caption.

RESPONSE TO REVIEWERS' COMMENTS

Reviewer #1 (Remarks to the Author):

In this revised manuscript the authors have successfully clarified all the questions raised, and added important information that strengthens their work. The expanded supplementary file offers very useful additional images and analysis, while the clarification of the EELS data and analysis, now into a simplified Figure 4, vastly improves the flow of the paper.

Similarly, points of technical details have been added to the methods, offering of more complete description of the work. The authors appear to have used the Atomap software package, and mention it in the context of the zigzag angle determination. A reference to the package repository (and access date), alongside with a reference to a paper describing this piece of software, which I believe are not currently included in the reference list, would be welcome.

Response: Thank you very much for your detailed feedback. We have added the missing information in the method part and references.

One final point may need to be briefly addressed (unlikely to require an additional round of reviews). The inclusion of simulated iCOM images is welcome, but it is puzzling to notice a clear mismatch in contrast between the Nd and NiO₂ columns comparing experiments and simulations. In figure 5, in the experimental data, they appear to have almost the same intensity, whereas the iCOM simulations suggest otherwise. I realize these are not meant to lead to quantitative contrast analysis, but the discrepancy is a little troubling - and may point to channeling effects or partial occupancies not fully captured by the models. A note of explanation on this observation is probably needed before final acceptance of the work - which I am otherwise happy to recommend for publication in Nature Communications.

Response: We are grateful for your insightful comments. We have carefully checked the parameters of the simulated images and find out the contrast difference with the experiment is due to the effects of vibrational scattering. The frozen phonon model has been used to simulate the effects of phonons on electron scattering during the simulation, obtaining the following simulated images (See Figure R1) comparable with the experimental results. We have modified Figure 5 and Figure S11 in the manuscript.

Figure R1 Comparison between the experimentally observed structure distortion and the relaxed structure model from DFT calculations. (a) Reconstructed iCoM image of domain A with clear polyhedral rotation and (b) the relaxed structure model with F intercalation at apical sites. (c) A simulated iCoM image of the structure in (b). The red dashed lines mark the anion sublattice distortion. (d) Reconstructed iCoM image of domain B showing a clear zigzag arrangement of Nd atoms and (e) the relaxed structure model with F intercalation at apical sites. (f) A simulated iCoM image of the structure in (e). The blue arrows indicate the displacement direction of the Nd atoms. The red dashed line marks the anion sublattice without distortion. Domain B is formed by a 90° rotation of domain A. (g) A relaxed structural model with two connected domains. (h) Reconstructed iCoM image of the domain interface and (i) the enlarged structural model. (j) A simulated iCoM image of the structure in (i). (k) Reconstructed iCoM image of the region with abnormal structure distortion and (l) the structure model from the DFT calculations with partial intercalation of F-ions at the basal sites. (m) A simulated iCoM image of the structure in (l).

Reviewer #3 (Remarks to the Author):

The authors have extensively addressed my comments with excellent thought from the first round of review. The manuscript is well-improved, also especially thanks to the comments from Reviewer 1. I have just a few minor comments left before publication in Nature Communications:

1. The large FOV HAADF-STEM that the authors provided are only shown in the review response but not in the edited manuscript. I would recommend showing these large FOV images in the Supplement as to provide useful sample images to the field. Also, there is significant structural long-range "bending" which looks almost periodic. Do the authors have any thoughts as to why the bending? This looks distinct from the disorder seen in doped NdNiO₂ on SrTiO₃ (no superlattice). Do the authors think the SrTiO₃ has some effect? I would expect the SrTiO₃ layers to prevent the formation of such bending features.

Response: We are thankful for your insightful comments. We have added the FOV HAADF-STEM in the supplemental information (See Figure S2). Regarding the bending of the superlattice sample, we have discussed the atomic structures at interfaces in the NdNiO₂/SrTiO₃ superlattice sample in our previous paper. (Ref., Nano Lett. 2023, 23, 8, 3291–3297) At the "bending" regions, there are atomic steps in the NdNiO₂/SrTiO₃ superlattice accompanying a step distribution of the apical oxygen at the interfaces, which is related to the elemental intermixing at the interfaces. It is probably related to the growth conditions of the sample and the surface roughness of the substrate. A similar "bending" is also visible in another reported result in another superlattice sample (Nd_{0.8}Sr_{0.2}NiO₂)₈/(SrTiO₃)₂]₁₀ (Ref. Nat. Commun. 2024, 15, 10215.) The disorder in the doped NdNiO₂ single film on grown on SrTiO₃ is due to the reduction process, which is different from the step interface in our superlattice sample.

In addition, at the interface region, the apical oxygen in the nickelate layer is not fully removed due to its connection with the SrTiO₃ layer. The SrTiO₃ layers have been observed to impede the formation of a disordered phase during the reduction process, manifesting longer reduction time beyond that of a single film.

2. The references on the Ni L₃/Ni L₂ edge intensity ratio were provided in the response letter but not cited in the updated manuscript.

Response: Thank you very much for your comments. We have added the description and references in the method part marked with green color in the revised manuscript.

3. Regarding the fluorinated NdNiO₂ vs NdNiO₃, the authors stated in their first response "However, the significant difference in the degree of fluorination for the two precursor phases makes it difficult to directly compare atomic scale TEM results between the samples." I would think that larger differences in the bulk-scale x-ray diffraction (which suggest that the NdNiO₂ is more fluorinated) would make it easier to identify what atomic- or meso-scale differences there are in the TEM. I would hesitate to hold publication for HAADF-STEM on the fluorinated NdNiO₃ samples but I do think such images would be very interesting and useful, even if exquisite atomic-scale analysis is not possible at this stage.

Response: We sincerely appreciate your insightful comments and apologize for any confusion caused by our previous response: "However, the significant difference in the degree of fluorination for the two precursor phases makes it difficult to directly compare atomic-scale TEM results between the samples." To clarify, variations in the degree of fluorination are likely to result in significant differences in atomic structure, whether in fluorinated NdNiO₂ or NdNiO₃ samples. The key point we intended to convey is

that, to meaningfully analyze the effects of fluorine intercalation on atomic structure, it is crucial to ensure an identical degree of fluorination in both cases. Given the differences in the degree of fluorination between NdNiO_3 and NdNiO_3 samples observed from XRD results, our previous statement was meant to emphasize this challenge.

In addition, we appreciate the point raised regarding the potential atomic- or meso-scale differences in TEM. To address the reviewer's concern, we have measured the HAADF images of the fluorinated NdNiO_3 sample, as shown in the Figure R2. Figure R2a shows the FOV HAADF-STEM of the fluorinated $8\text{NdNiO}_3/4\text{SrTiO}_3$ superlattice sample and the corresponding enlarged HAADF images of b and c show the atomic structure of the sample. In Figures R2b and c, the zigzag arrangement of Nd columns is visible and widely distributed in $\text{NdNiO}_{x}\text{F}_y$ layers. Notably, the direction of the zigzag arrangement of Nd is vertical to the $8\text{NdNiO}_3/4\text{SrTiO}_3$ interfaces, while it is parallel to the interfaces in the fluorinated $8\text{NdNiO}_2/4\text{SrTiO}_3$ superlattice sample. Besides, there is no clear domain formation in the $\text{NdNiO}_{x}\text{F}_y$ layers, which is likely to be closely related to the extent of fluorination.

However, distinguishing the effects of the fluorine intercalation method and the degree of fluorination on the atomic structure remains challenging in this experiment. We appreciate the reviewer's comments highlighting this complexity and consider it a promising direction for future experimental studies to investigate the differences between these phases in greater detail.

Figure R2 (a) Overview HAADF-STEM of the fluorinated $8\text{NdNiO}_3/4\text{SrTiO}_3$ superlattice sample, and the enlarged HAADF images of (b) and (c). The magnified regions are indicated by the red dashed boxes.